# FAST TOPOLOGICAL CLUSTERING
# WITH WASSERSTEIN DISTANCE

**Tananun Songdechakraiwut**[*]
Department of Electrical and Computer Engineering
University of Wisconsin–Madison, USA
`songdechakra@wisc.edu`

**Bryan M. Krause & Matthew I. Banks**
Department of Anesthesiology
Department of Neuroscience
University of Wisconsin–Madison, USA

**Kirill V. Nourski**
Department of Neurosurgery
Iowa Neuroscience Institute
University of Iowa, USA

**Barry D. Van Veen**
Department of Electrical and Computer Engineering
University of Wisconsin–Madison, USA

## ABSTRACT

The topological patterns exhibited by many real-world networks motivate the development of topology-based methods for assessing the similarity of networks. However, extracting topological structure is difficult, especially for large and dense networks whose node degrees range over multiple orders of magnitude. In this paper, we propose a novel and computationally practical *topological clustering* method that clusters complex networks with intricate topology using principled theory from persistent homology and optimal transport. Such networks are aggregated into clusters through a centroid-based clustering strategy based on both their topological and geometric structure, preserving correspondence between nodes in different networks. The notions of topological proximity and centroid are characterized using a novel and efficient approach to computation of the Wasserstein distance and barycenter for persistence barcodes associated with connected components and cycles. The proposed method is demonstrated to be effective using both simulated networks and measured functional brain networks.

## 1 INTRODUCTION

Network models are extremely useful representations for complex data. Significant attention has been given to cluster analysis within a single network, such as detecting community structure (Newman, 2006; Rohe et al., 2011; Yin et al., 2017). Less attention has been given to clustering of collections of network representations. Clustering approaches typically group similar networks based on comparisons of edge weights (Xu & Wunsch, 2005), not topology. Assessing similarity of networks based on topological structure offers the potential for new insight, given the inherent topological patterns exhibited by most real-world networks. However, extracting meaningful network topology is a very difficult task, especially for large and dense networks whose node degrees range over multiple orders of magnitude (Barrat et al., 2004; Bullmore & Sporns, 2009; Honey et al., 2007).

Persistent homology (Barannikov, 1994; Edelsbrunner et al., 2000) has recently emerged as a powerful tool for understanding, characterizing and quantifying complex networks (Songdechakraiwut et al., 2021). Persistent homology represents a network using topological features such as connected components and cycles. Many networks naturally divide into modules or connected components (Bullmore & Sporns, 2009; Honey et al., 2007). Similarly, cycles are ubiquitous and are often used to describe information propagation, robustness and feedback mechanisms (Kwon & Cho, 2007; Lind et al., 2005). Effective use of such topological descriptors requires a notion of proximity that quantifies the similarity between persistence barcodes, a convenient representation for connected components and cycles (Ghrist, 2008). Wasserstein distance, which measures the minimal effort to modify one persistence barcode to another (Rabin et al., 2011), is an excellent choice due to its

---

[*]Code for topological clustering is available at `https://github.com/topolearn`.

appealing geometric properties (Staerman et al., 2021) and its effectiveness shown in many machine learning applications (Kolouri et al., 2017; Mi et al., 2018; Solomon et al., 2015). Importantly, Wasserstein distance can be used to interpolate networks while preserving topological structure (Songdechakraiwut et al., 2021), and the mean under the Wasserstein distance, known as Wasserstein barycenter (Agueh & Carlier, 2011), can be viewed as the topological centroid of a set of networks.

The high cost of computing persistence barcodes, Wasserstein distance and the Wasserstein barycenter limit their applications to small scale problems, see, e.g., (Clough et al., 2020; Hu et al., 2019; Kolouri et al., 2017; Mi et al., 2018). Although approximation algorithms have been developed (Cuturi, 2013; Cuturi & Doucet, 2014; Lacombe et al., 2018; Li et al., 2020; Solomon et al., 2015; Vidal et al., 2019; Xie et al., 2020; Ye et al., 2017), it is unclear whether these approximations are effective for clustering complex networks as they inevitably limit sensitivity to subtle topological features. Indeed, more and more studies, see, e.g., (Robins & Turner, 2016; Xia & Wei, 2014) have demonstrated that such subtle topological patterns are important for the characterization of complex networks, suggesting these approximation algorithms are undesirable.

Recently, it was shown that the Wasserstein distance and barycenter for network graphs have closed-form solutions that can be computed exactly and efficiently (Songdechakraiwut et al., 2021) because the persistence barcodes are inherently one dimensional. Motivated by this result, we present a novel and computationally practical *topological clustering* method that clusters complex networks of the same size with intricate topological characteristics. Topological information alone is effective at clustering networks when there is no correspondence between nodes in different networks. However, when networks have meaningful node correspondence, we perform cluster analysis using combined topological and geometric information to preserve node correspondence. Statistical validation based on ground truth information is used to demonstrate the effectiveness of our method when discriminating subtle topological features in simulated networks. The method is further illustrated by clustering measured functional brain networks associated with different levels of arousal during administration of general anesthesia. Our proposed method outperforms other clustering approaches in both the simulated and measured data.

The paper is organized as follows. Background on our one-dimensional representation of persistence barcodes is given in section 2, while section 3 presents our topological clustering method. In sections 4 and 5, we compare the performance of our method to several baseline algorithms using simulated and measured networks, and conclude the paper with a brief discussion of the potential impact of this work.

## 2 ONE-DIMENSIONAL PERSISTENCE BARCODES

### 2.1 GRAPH FILTRATION

Consider a network represented as a weighted graph $G = (V, \boldsymbol{w})$ comprising a set of nodes $V$ with symmetric adjacency matrix $\boldsymbol{w} = (w_{ij})$, with edge weight $w_{ij}$ representing the relationship between node $i$ and node $j$. The number of nodes is denoted as $|V|$. The binary graph $G_\epsilon = (V, \boldsymbol{w}_\epsilon)$ of $G$ is defined as a graph consisting of the node set $V$ and binary edge weights $w_{\epsilon,ij} = 1$ if $w_{ij} > \epsilon$ and $w_{\epsilon,ij} = 0$ otherwise. We view the binary network $G_\epsilon$ as a 1-skeleton (Munkres, 2018), a simplicial complex comprising only nodes and edges. In the 1-skeleton, there are two types of topological features: connected components (0-dimensional topological features) and cycles (1-dimensional topological features). There are no topological features of higher dimensions in the 1-skeleton, in contrast to well-known Rips (Ghrist, 2008) and clique complexes (Otter et al., 2017). The number of connected components and the number of cycles in the binary network are referred to as the 0-th Betti number $\beta_0(G_\epsilon)$ and the 1-st Betti number $\beta_1(G_\epsilon)$, respectively. A *graph filtration* of $G$ is defined as a collection of nested binary networks (Lee et al., 2012):

$$G_{\epsilon_0} \supseteq G_{\epsilon_1} \supseteq \cdots \supseteq G_{\epsilon_k},$$

where $\epsilon_0 \leq \epsilon_1 \leq \cdots \leq \epsilon_k$ are filtration values. As $\epsilon$ increases, more and more edges are removed from the network $G$ since we threshold the edge weights at higher connectivity. For instance, $G_{-\infty}$ has each pair of nodes connected by an edge and thus is a complete graph consisting of a single connected component, while $G_\infty$ has no edges and represents the node set. Figure 1 illustrates the graph filtration of a four-node network and the corresponding Betti numbers. Note that other

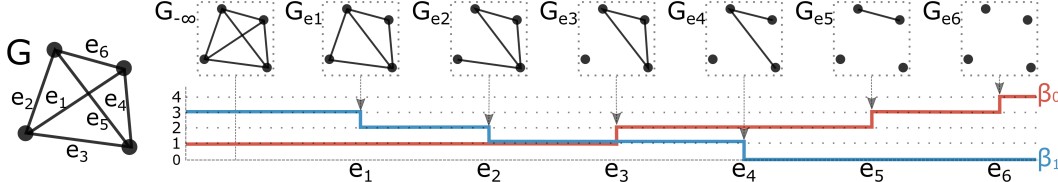

Figure 1: Graph filtration of four-node network $G$. As the filtration value increases, the number of connected components $\beta_0$ monotonically increases while the number of cycles $\beta_1$ monotonically decreases. Connected components are born at the edge weights $e_3, e_5, e_6$ while cycles die at $e_1, e_2, e_4$.

filtrations for analyzing graphs have been proposed, including use of descriptor functions such as heat kernels (Carrière et al., 2020) and task-specific learning (Hofer et al., 2020).

### 2.2 BIRTH-DEATH DECOMPOSITION

Persistent homology keeps track of birth and death of connected components and cycles over filtration values $\epsilon$ to deduce their *persistence*, i.e., the lifetime from their birth to death over $\epsilon$. The persistence is represented as a *persistence barcode* $PB(G)$ comprising intervals $[b_i, d_i]$ representing the lifetime of a connected component or a cycle that appears at the filtration value $b_i$ and vanishes at $d_i$.

In the edge-weight threshold graph filtration defined in Section 2.1, connected components are born and cycles die as the filtration value increases (Songdechakraiwut et al., 2021). Specifically, $\beta_0$ is monotonically increasing from $\beta_0(G_{-\infty}) = 1$ to $\beta_0(G_\infty) = |V|$. There are $\beta_0(G_\infty) - \beta_0(G_{-\infty}) = |V| - 1$ connected components that are born over the filtration. Connected components will never die once they are born, implying that every connected component has a death value at $\infty$. Thus, we can represent their persistence as a collection of finite birth values $B(G) = \{b_i\}_{i=1}^{|V|-1}$. On the other hand, $G_{-\infty}$ is a complete graph containing all possible cycles; thus, all cycles have birth values at $-\infty$. Again, we can represent the persistence of the cycles as a collection of finite death values $D(G) = \{d_i\}$. How many cycles are there? Since the deletion of an edge $w_{ij}$ must result in either the birth of a connected component or the death of a cycle, every edge weight must be in either $B(G)$ or $D(G)$. Thus, the edge weight set $W = \{w_{ij} | i > j\}$ decomposes into the collection of birth values $B(G)$ and the collection of death values $D(G)$. Since $G_{-\infty}$ is a complete graph with $\frac{|V|(|V|-1)}{2}$ edge weights and $|V| - 1$ of these weights are associated with the birth of connected components, the number of cycles in $G_{-\infty}$ is thus equal to $\frac{|V|(|V|-1)}{2} - (|V| - 1) = 1 + \frac{|V|(|V|-3)}{2}$. In the example of Figure 1, we have $B(G) = \{e_3, e_5, e_6\}$ and $D(G) = \{e_1, e_2, e_4\}$. Other graph filtrations (Carrière et al., 2020; Hofer et al., 2020) do not necessarily share this monotonicity property and consequently one-dimensional barcode representations are not applicable.

Finding the birth values in $B(G)$ is *equivalent* to finding edge weights comprising the *maximum spanning tree* of $G$ and can be done using well-known methods such as Prim's and Kruskal's algorithms (Lee et al., 2012). Once $B(G)$ is known, $D(G)$ is simply given as the remaining edge weights. Finding $B(G)$ and $D(G)$ requires only $O(n \log n)$ operations, where $n$ is the number of edges in the network, and thus is extremely computationally efficient.

## 3 CLUSTERING METHOD

### 3.1 TOPOLOGICAL DISTANCE SIMPLIFICATION

Use of edge-weight threshold filtration and limiting consideration to connected components and cycles as topological features results in significant simplification of the 2-Wasserstein distance (Rabin et al., 2011) between barcode descriptors (Cohen-Steiner et al., 2010) of networks as follows. Let $G$ and $H$ be two given networks that have the same number of nodes. The *topological distance* $d_{top}(G, H)$ is defined as the optimal matching cost:

$$\left( \min_\tau \sum_{p \in PB(G)} ||p - \tau(p)||^2 \right)^{\frac{1}{2}} = \left( \min_\tau \sum_{p = [b_p, d_p] \in PB(G)} \left[ b_p - b_{\tau(p)} \right]^2 + \left[ d_p - d_{\tau(p)} \right]^2 \right)^{\frac{1}{2}}, \quad (1)$$

where the optimization is over all possible bijections $\tau$ from barcode $PB(G)$ to barcode $PB(H)$. Intuitively, we can think of each interval $[b_i, d_i]$ as a point $(b_i, d_i)$ in a 2-dimensional plane. The topological distance measures the minimal amount of work to move points in $PB(G)$ to $PB(H)$. Note this alternative representation of points in the plane is equivalent to the persistence barcode and called the *persistence diagram* (Edelsbrunner & Harer, 2008). Moving a connected component point $(b_i, \infty)$ to a cycle point $(-\infty, d_j)$ or vice versa takes an infinitely large amount of work. Thus, we only need to optimize over bijections that match the same type of topological features. Subsequently, we can equivalently rewrite $d_{top}$ in terms of $B(G), D(G), B(H)$ and $D(H)$ as

$$d_{top}(G, H) = \left( \min_{\tau_0} \sum_{b \in B(G)} \left[ b - \tau_0(b) \right]^2 + \min_{\tau_1} \sum_{d \in D(G)} \left[ d - \tau_1(d) \right]^2 \right)^{\frac{1}{2}}, \tag{2}$$

where $\tau_0$ is a bijection from $B(G)$ to $B(H)$ and $\tau_1$ is a bijection from $D(G)$ to $D(H)$. The first term matches connected components to connected components and the second term matches cycles to cycles. Matching each type of topological feature separately is commonly done in medical imaging and machine learning studies (Clough et al., 2020; Hu et al., 2019). The topological distance $d_{top}$ has a closed-form solution that allows for efficient computation as follows (Songdechakraiwut et al., 2021).

$$d_{top}(G, H) = \left( \sum_{b \in B(G)} \left[ b - \tau_0^*(b) \right]^2 + \sum_{d \in D(G)} \left[ d - \tau_1^*(d) \right]^2 \right)^{\frac{1}{2}}, \tag{3}$$

where $\tau_0^*$ maps the $l$-th smallest birth value in $B(G)$ to the $l$-th smallest birth value in $B(H)$ and $\tau_1^*$ maps the $l$-th smallest death value in $D(G)$ to the $l$-th smallest death value in $D(H)$ for all $l$.

A proof is provided in Appendix A. As a result, the optimal matching cost can be computed quickly and efficiently by sorting birth and death values, and matching them in order. The computational cost of evaluating $d_{top}$ is $O(n \log n)$, where $n$ is the number of edges in networks.

## 3.2 TOPOLOGICAL CLUSTERING

Let $G = (V, \boldsymbol{w})$ and $H = (V, \boldsymbol{u})$ be two networks. We define the network dissimilarity $d_{net}^2$ between $G$ and $H$ as a weighted sum of the squared geometric distance and the squared topological distance:

$$d_{net}^2(G, H) = (1 - \lambda) \sum_i \sum_{j > i} \left( w_{ij} - u_{ij} \right)^2 + \lambda d_{top}^2(G, H), \tag{4}$$

where $\lambda \in [0, 1]$ controls the relative weight between the geometric and topological terms. The geometric distance measures the node-by-node dissimilarity in the networks that is not captured by topology alone and is helpful when node identity is meaningful, such as in neuroscience applications. Given observed networks with identical node sets, the goal is to partition the networks into $k$ clusters $\mathcal{C} = \{C_h\}_{h=1}^k$ with corresponding cluster centroids or representatives $\mathcal{M} = \{M_h\}_{h=1}^k$ such that the sum of the network dissimilarities $d_{net}^2$ from the networks to their representatives is minimized, i.e.,

$$\min_{\mathcal{C}, \mathcal{M}} L(\mathcal{C}, \mathcal{M}) = \min_{\mathcal{C}, \mathcal{M}} \sum_{h=1}^k \sum_{G \in C_h} d_{net}^2(M_h, G). \tag{5}$$

The *topological clustering* formulation given in (5) suggests a natural iterative relocation algorithm using coordinate descent (Banerjee et al., 2005). In particular, the algorithm alternates between two steps: an assignment step and a re-estimation step. In the assignment step, $L$ is minimized with respect to $\mathcal{C}$ while holding $\mathcal{M}$ fixed. Minimization of $L$ is achieved simply by assigning each observed network to the cluster whose representative $M_h$ is the nearest in terms of the criterion $d_{net}^2$. In the re-estimation step, the algorithm minimizes $L$ with respect to $\mathcal{M}$ while holding $\mathcal{C}$ fixed. In this case, $L$ is minimized by re-estimating the representatives for each individual cluster:

$$\min_{\mathcal{M}} \sum_{h=1}^k \sum_{G \in C_h} d_{net}^2(M_h, G) = \sum_{h=1}^k \min_{M_h} \sum_{G \in C_h} d_{net}^2(M_h, G). \tag{6}$$

We will consider solving the objective function given in (6) for $\lambda = 0, \lambda = 1$ and $\lambda \in (0, 1)$.

$\lambda = 0$ describes conventional edge clustering (MacQueen, 1967) since the topological term is excluded.

$\lambda = 1$ describes clustering based on pure topology and ignores correspondence of edge weights. This case is potentially useful for clustering networks whose node sets are not identical or of different size. Each representative $M_h$ of cluster $C_h$ minimizes the sum of the squared topological distances, i.e.,

$$\min_{M_h} \sum_{G \in C_h} d_{top}^2(M_h, G) = \min_{B(M_h), D(M_h)} \sum_{G \in C_h} \Big( \sum_{b \in B(M_h)} \big[ b - \tau_0^*(b) \big]^2 + \sum_{d \in D(M_h)} \big[ d - \tau_1^*(d) \big]^2 \Big). \tag{7}$$

Thus, we only need to optimize over the topology of the network, i.e., $B(M_h)$ and $D(M_h)$, instead of the original network $M_h$ itself. The topology solving (7) is the *topological centroid* of networks in cluster $C_h$. Interestingly, the topological centroid has a closed-form solution and can be calculated analytically as follows.

**Lemma 1.** *Let $G_1, ..., G_n$ be $n$ networks each with $m$ nodes. Let $B(G_i) : b_{i,1} \leq \cdots \leq b_{i,m-1}$ and $D(G_i) : d_{i,1} \leq \cdots \leq d_{i,1+m(m-3)/2}$ be the topology of $G_i$. It follows that the $l$-th smallest birth value of the topological centroid of the $n$ networks is given by the mean of all the $l$-th smallest birth values of such networks, i.e., $\sum_{i=1}^n b_{i,l}/n$. Similarly, the $l$-th smallest death value of the topological centroid is given by $\sum_{i=1}^n d_{i,l}/n$.*

Since Eq. (7) is quadratic, Lemma 1 can be proved by setting its derivative equal to zero. The complete proof is given in Appendix A. The results in (Songdechakraiwut & Chung, 2020; Songdechakraiwut et al., 2022) may be used to find the centroid of different size networks.

For the most general case considering both correspondence of edge weights and topology, i.e., when $\lambda \in (0, 1)$, we can optimize the objective function given in (6) by gradient descent. Let $H = (V, \boldsymbol{u})$ be a cluster representative being estimated given $C_h$. The gradient of the squared topological distance $\nabla_H d_{top}^2(H, G)$ with respect to edge weights $\boldsymbol{u} = (u_{ij})$ is given as a gradient matrix whose $ij$-th entry is

$$\frac{\partial d_{top}^2(H, G)}{\partial u_{ij}} = \begin{cases} 2\big[u_{ij} - \tau_0^*(u_{ij})\big] & \text{if } u_{ij} \in B(H); \\ 2\big[u_{ij} - \tau_1^*(u_{ij})\big] & \text{if } u_{ij} \in D(H). \end{cases} \tag{8}$$

This follows because the edge weight set decomposes into the collection of births and the collection of deaths. Intuitively, by slightly adjusting the edge weight $u_{ij}$, we have the slight adjustment of either a birth value in $B(H)$ or a death value in $D(H)$, which slightly changes the topology of the network $H$. The gradient computation consists of computing persistence barcodes and finding the optimal matching using the closed-form solution given in (3), requiring $O(n \log n)$ operations, where $n$ is the number of edges in networks.

Evaluating the gradient for (6) requires computing the gradients of all the observed networks. This can be computationally demanding when the size of a dataset is large. However, an equivalent minimization problem that allows faster computation is possible using the following result:

**Lemma 2.**

$$\sum_{h=1}^k \min_{H_h} \sum_{G \in C_h} d_{net}^2(H_h, G) = \sum_{h=1}^k \min_{H_h} \Big( (1 - \lambda) \sum_i \sum_{j > i} \big( u_{h,ij} - \overline{w}_{h,ij} \big)^2 + \lambda d_{top}^2 \big( H_h, \widehat{M}_{top,h} \big) \Big), \tag{9}$$

*where $\overline{\boldsymbol{w}}_{\boldsymbol{h}} = (\overline{w}_{h,ij})$ are edge weights in the sample mean network $\overline{M}_h = (V, \overline{\boldsymbol{w}}_{\boldsymbol{h}})$ of cluster $C_h$, and $\widehat{M}_{top,h}$ is the topological centroid of networks in cluster $C_h$.*

Thus, instead of computing the gradient for every network in the set, it is sufficient to compute the gradient at the cluster sample means $\overline{M}_h$ and topological centroids $\widehat{M}_{top,h}$. Hence, one only needs to perform *topological interpolation* between the sample mean network $\overline{M}_h$ and the topological centroid $\widehat{M}_{top,h}$ of each cluster. That is, the optimal representative is the one whose geometric location is close to $\overline{M}_h$ and topology is similar to $\widehat{M}_{top,h}$. At each current iteration, we take a step in the direction of negative gradient with respect to an updated $H_h$ from the previous iteration. Note that the proposed method constrains the search of cluster centroids to a space of meaningful networks through topological interpolation. In contrast, the sample mean, such as would be used in a naive application of $k$-means, does not necessarily represent a meaningful network.

Furthermore, we have the following theorem:

**Theorem 1.** *The topological clustering algorithm monotonically decreases $L$ in (5) and terminates in a finite number of steps at a locally optimal partition.*

The proofs for Lemma 2 and Theorem 1 are provided in Appendix A.

## 4 VALIDATION USING SIMULATED NETWORKS

Simulated networks of different topological structure are used to evaluate the clustering performance of the proposed approach relative to that of seven other methods. We use the term *topology* to denote the proposed approach with $\lambda > 0$. Clustering using the special case of $\lambda = 0$ is termed the *edge-weight* method. Recall the graph filtration defined in section 2.2 decomposes the edge weights into two groups of birth and death values corresponding to connected components and cycles. In order to assess the impact of separating edge weights into birth and death sets, we evaluate an ad hoc method called *sort* that applies $k$-means to feature vectors obtained by simply sorting edge weights.

In addition, we evaluate several clustering algorithms that utilize network representations. Two persistent homology clustering methods based on conventional two-dimensional barcodes (Otter et al., 2017) are evaluated. The *Wasserstein* method performs clustering using $k$-medoids based on Wasserstein distance between two-dimensional barcodes for connected components and cycles. The computational complexity of the *Wasserstein* method is managed by utilizing an approximation algorithm (Lacombe et al., 2018) to compute the Wasserstein distance. A Wasserstein barycenter clustering algorithm called *Bregman Alternating Direction Method of Multipliers* (B-ADMM) (Ye et al., 2017) is used to cluster two-dimensional barcodes for cycles. In order to make B-ADMM computationally tractable, each barcode is limited to no more than the 50 most persistent cycles.

The *Net Stats* clustering approach uses $k$-means to cluster three-dimensional feature vectors composed of the following network summary statistics: (1) average weighted degree over all nodes (Rubinov & Sporns, 2010), (2) average clustering coefficient over all nodes (Rubinov & Sporns, 2010) and (3) modularity (Newman, 2006; Reichardt & Bornholdt, 2006).

Lastly, we evaluate two graph kernel based clustering approaches using kernel $k$-means (Dhillon et al., 2004): the *GraphHopper* kernel (Feragen et al., 2013) and the *propagation* kernel (Neumann et al., 2016). Supplementary description and implementation details of the candidate methods are provided in Appendix B.

Initial clusters for all methods are selected at random.

**Modular network structure** Random modular networks $\mathcal{X}_i$ are simulated with $|V|$ nodes and $m$ modules such that the nodes are evenly distributed among modules. Figure 2 displays modular networks with $|V| = 60$ nodes and $m = 5$ modules such that $|V|/m = 12$ nodes are in each module. Edges connecting two nodes within the same module are assigned a random weight following a normal distribution $\mathcal{N}(\mu, \sigma^2)$ with probability $r$ or otherwise Gaussian noise $\mathcal{N}(0, \sigma^2)$ with probability $1 - r$. On the other hand, edges connecting nodes in different modules have probability $1 - r$ of being $\mathcal{N}(\mu, \sigma^2)$ and probability $r$ of being $\mathcal{N}(0, \sigma^2)$. The modular structure becomes more pronounced as the within-module connection probability $r$ increases. Any negative edge weights are set to zero. This procedure yields random networks $\mathcal{X}_i$ that exhibit topological connectedness. We use $\mu = 1$ and $\sigma = 0.5$ universally throughout the study.

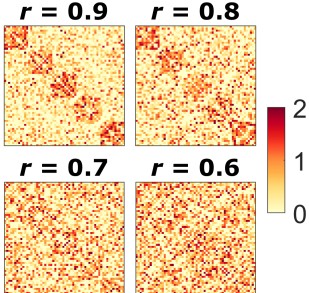

Figure 2: Example networks with $|V| = 60$ nodes and $m = 5$ modules exhibit different within-module connection probabilities $r = 0.9, 0.8, 0.7$ and $0.6$.

**Simulation** Three groups of modular networks $L_1 = \{\mathcal{X}_i\}_{i=1}^{20}$, $L_2 = \{\mathcal{X}_i\}_{i=21}^{40}$ and $L_3 = \{\mathcal{X}_i\}_{i=41}^{60}$ corresponding to $m = 2, 3$ and $5$ modules, respectively, are simulated. This results in 60 networks in the dataset, each of which has a group label $L_1, L_2$ or $L_3$. We consider $r = 0.9, 0.8, 0.7$ and $0.6$ to vary the strength of the modular structure, as illustrated in Figure 2.

The dataset is partitioned into three clusters $C_1, C_2$ and $C_3$ using the candidate algorithms. Clustering performance is then evaluated by first assigning each cluster to the group label that is most frequent in that cluster and then calculating the accuracy statistic $s$ as the fraction of correctly labeled networks,

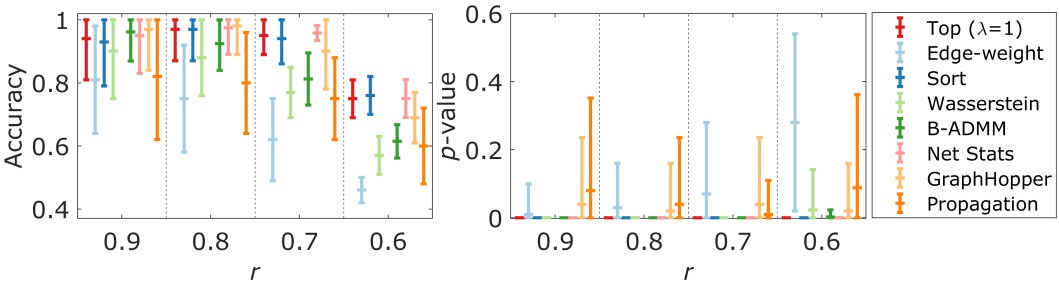

Figure 3: Clustering performance comparison for the dataset of simulated networks with $|V| = 60$ nodes and $m = 2, 3, 5$ modules with respect to average accuracy (left) and average $p$-values (right). Results for within-module connection probabilities $r = 0.6, 0.7, 0.8$ and $0.9$ are shown. Data points (middle horizontal lines) indicate the averages over results for 100 different initial conditions, and vertical error bars indicate standard deviations.

i.e., $s = \frac{1}{60} \sum_{i=1}^{3} \max_j \{|C_i \cap L_j|\}$, where $|C_i \cap L_j|$ denotes the number of common networks in both $C_i$ and $L_j$. Note this evaluation of clustering performance is called *purity*, which not only is transparent and interpretable but also works well in this simulation study where the number and size of clusters are small and balanced, respectively (Manning et al., 2008). Since the distribution of the accuracy $s$ is unknown, a permutation test is used to determine the empirical distribution under the null hypothesis that sample networks and their group labels are independent (Ojala & Garriga, 2010). The empirical distribution is calculated by repeatedly shuffling the group labels and then re-computing the corresponding accuracy for one million random permutations. By comparing the observed accuracy to this empirical distribution, we can determine the statistical significance of the clustering performance. The $p$-value is calculated as the fraction of permutations that give accuracy higher than the observed accuracy $s$. The average $p$-value and average accuracy across 100 different initial assignments are reported.

**Results** Figure 3 indicates that all methods considered perform relatively well with pronounced modularity ($r = 0.9$). As the modularity strength diminishes with decreasing $r$, clustering performance also decreases. The proposed topological clustering algorithm ($\lambda = 1$) performs relatively well for the more nuanced modularity associated with $r = 0.7$ and $0.6$. Since the dataset is purposefully generated to exhibit dependency between the sample networks and their group labels, the $p$-value indicates the degree of statistical significance to which structure is differentiated (Ojala & Garriga, 2010). Small $p$-values indicate the algorithm is able to differentiate network structure. The proposed method has very low $p$-values indicating that its improved accuracy over the baseline methods is significant. The *Wasserstein* and B-ADMM

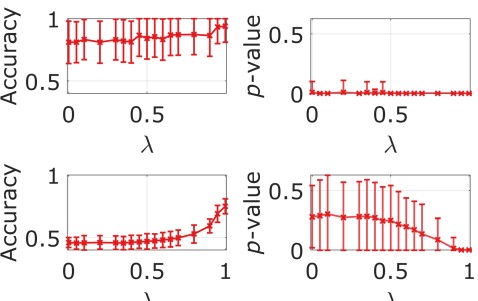

Figure 4: Clustering performance for simulated networks with $|V| = 60$ nodes and $m = 2, 3, 5$ modules as a function of $\lambda$ for within-module connection probabilities $r = 0.9$ (top row) and $r = 0.6$ (bottom row).

methods require very high computation costs for the conventional two-dimensional barcodes (Otter et al., 2017), and the approximations of Wasserstein distance (Lacombe et al., 2018) and barycenter (Ye et al., 2017). The ad hoc *sort* method performs nearly identically to the topology approach, showing that the separation into births of connected components and deaths of cycles has no apparent impact for these simulated networks with densely connected structure. However, we hypothesize the separation will impact performance when networks are sparse. Supplementary experimental results including an additional set of module sizes $m = 2, 5$ and $10$ are provided in Appendix B.

Figure 4 displays topological clustering performance as a function of $\lambda$ for the strongest ($r = 0.9$) and weakest ($r = 0.6$) degree of modularity. The performance is not very sensitive to the value of $\lambda$. When the within-module connection probability is small ($r = 0.6$), increasing the weight of the topological distance by increasing $\lambda$ results in the best performance. We hypothesize that in this case the relatively strong random nature of the edge weights introduces noise into the geometric term that

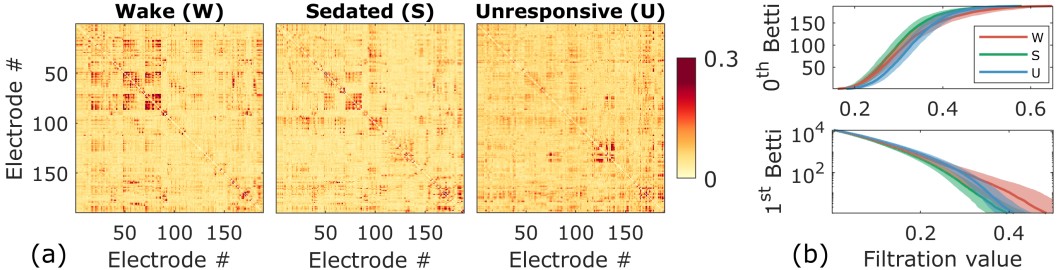

Figure 5: Representative data from a single subject (ID R376). For this subject there are 36 measured networks in each of three conditions: wake, sedated, and unresponsive states for a total of 108 networks. (a) Sample mean networks during wake, sedated and unresponsive states of the subject computed using ground truth labels. (b) Betti plots based on ground truth labels. Thick lines represent topological centroids. Shaded areas around the centroids represent standard deviation.

hinders performance as $\lambda$ decreases. Conversely, the topological distance appears less sensitive to this particular edge weight noise, resulting in the best performance when $\lambda = 1$.

## 5 APPLICATION TO FUNCTIONAL BRAIN NETWORKS

**Dataset** We evaluate our method using an extended brain network dataset from the anesthesia study reported by Banks et al. (2020) (see Appendix C). The measured brain networks are based on alpha band (8-12 Hz) weighted phase lag index (Vinck et al., 2011) applied to 10-second segments of resting state intracranial electroencephalography recordings from eleven neurosurgical patients administered increasing doses of the general anesthetic propofol just prior to surgery. The network size varies from 89 to 199 nodes while the number of networks (10-second segments) per subject varies from 71 to 119. Each segment is labeled as one of the three arousal states: pre-drug wake (W), sedated/responsive (S), or unresponsive (U); these labels are used as ground truth in the cluster analysis. Figure 5 illustrates sample mean networks and Betti plots describing topology for a representative subject.

**Performance evaluation** We apply the adjusted Rand index (ARI) (Hubert & Arabie, 1985) to compare clustering performance against ground truth. Lower scores indicate less similarity while higher scores show higher similarity between estimated and ground-truth clusters. Perfect agreement is scored as 1.0. For each subject, we calculate these performance metrics by running the algorithm for 100 different initial conditions, resulting in 100 scores which are then averaged. We also average across subjects ($11 \times 100$ scores) to obtain a final overall score, which describes the overall output clusters across trials and subjects. We calculate average confusion matrices for each method by assigning each cluster to the state that is most frequent in that cluster.

**Cluster analysis** All baselines used in the simulation study are evaluated on the brain network dataset. In addition, we consider clustering using $k$-medoids and three previously proposed network distance measures for brain network analyses: *Gromov-Hausdorff* (GH) (Lee et al., 2012); *Kolmogorov-Smirnov* (KS) (Chung et al., 2019); and the *spectral* norm (Banks et al., 2020). Supplementary description of the three methods is provided in Appendix B. Initial clusters are selected at random for all methods. Figure 6 reports the ARI of the topological approach for multiple predefined $\lambda$'s including $\lambda = 0$ and $\lambda = 1$. The relative performance of topological clustering is reported in Figures 7 and 8 assuming $\lambda = 0.5$, which results in equal weighting of topological and geometric distances.

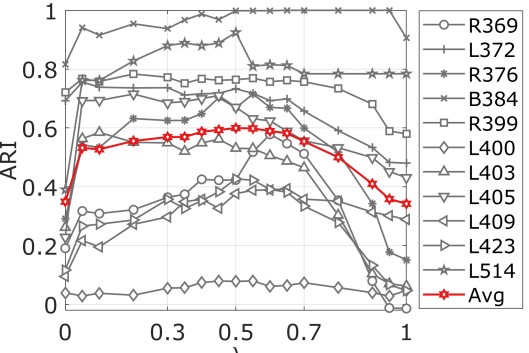

Figure 6: Average ARI per subject (gray lines) and across all eleven subjects (red line) as a function of $\lambda$.

**Results** Figure 6 indicates that the performance on the brain network data set varies significantly across subjects, but in general varies stably with $\lambda$. The best performance occurs with $\lambda$ between 0.4

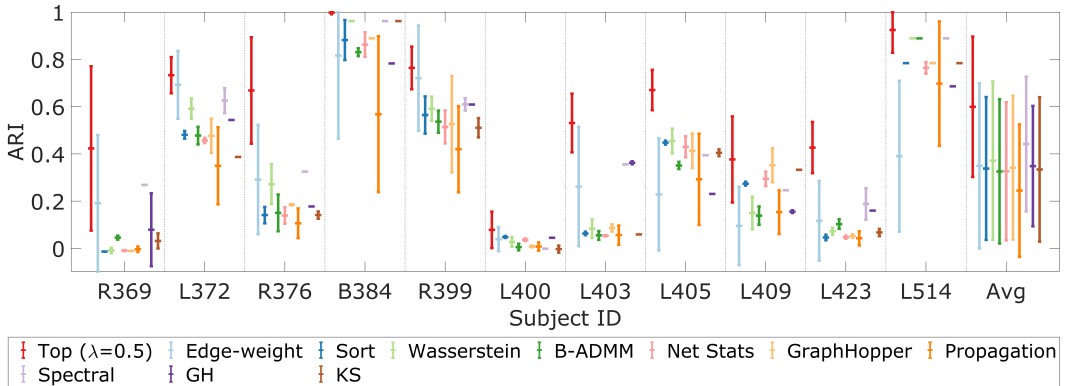

Figure 7: ARI performance for the brain network dataset. Data points (middle horizontal lines) indicate averages over 100 random initial conditions, and error bars indicate standard deviations.

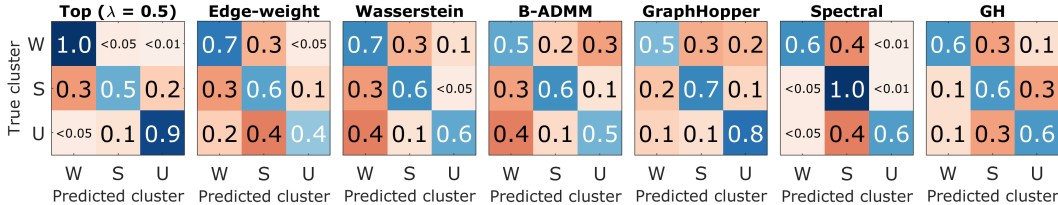

Figure 8: Average confusion matrices over 100 random initializations for select methods for subject R376 (see Figure 5). Results for the other methods are provided in Appendix B.

and 0.7, suggesting that a combination of geometric and topological distances gives the best result and that performance is not sensitive to the exact value chosen for $\lambda$. Note that different regions of the brain are functionally differentiated, so it is plausible that topological changes due to changing arousal level vary with location, giving geometric distance a role in the clustering of brain networks. Prior expectations of this sort can be used to determine whether to cluster based only on topology ($\lambda = 1$) or a combination of geometry and topology.

Figure 7 compares the ARI performance metric for individual subjects and the combination of all the subjects. All methods demonstrate significant variability in individual subject performance. This is expected due to the variations in effective signal to noise ratio, the number of nodes, and the number of networks in each underlying state. Consequently the combined performance across subjects has high variability. The topological method with $\lambda = 0.5$ performs relatively well across all subjects. Figure 8 illustrates that the proposed approach is particularly effective at separating wake (W) and unresponsive (U) states. Transitioning from the wake state to the unresponsive state results in dramatic changes in brain connectivity (Banks et al., 2020). The majority of errors from the proposed topological clustering approach are associated with the natural overlap between wake and sedated states (S). The sedated brain, in which subjects have been administered propofol but are still conscious, is expected to be more like the wake brain than the unresponsive brain. Thus, errors are expected to be more likely in differentiating sedated and wake states than sedated and unresponsive states. This suggests that the types of errors observed in the proposed method are consistent with prior biological expectations.

**Impact** The demonstrated effectiveness and computational elegance of our approach to clustering networks based on topological similarity will have a high impact on the analysis of large and complex network representations. In the study of brain networks, algorithms that can demonstrate correlates of behavioral states are of considerable interest. The derived biomarkers of changes in arousal state presented here demonstrate potential for addressing the important clinical problem of passively assessing arousal state in clinical settings, e.g., monitoring depth of anesthesia and in establishing diagnosis and prognosis for patients with traumatic brain injury and other disorders of consciousness. More broadly, the algorithm presented here will contribute to elucidating the neural basis of consciousness, one of the most important open problems in biomedical science.

ACKNOWLEDGMENTS

This work is supported in part by the National Institute of General Medical Sciences under award R01 GM109086, and the Lynn H. Matthias Professorship from the University of Wisconsin.

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

APPENDIX

## A PROOFS

### A.1 CLOSED-FORM SOLUTION OF TOPOLOGICAL DISTANCE

The topological distance $d_{top}$ has the closed-form solution as follows

$$d_{top}(G, H) = \Big( \sum_{b \in B(G)} \big[b - \tau_0^*(b)\big]^2 + \sum_{d \in D(G)} \big[d - \tau_1^*(d)\big]^2 \Big)^{\frac{1}{2}},$$

where $\tau_0^*$ maps the $l$-th smallest birth value in $B(G)$ to the $l$-th smallest birth value in $B(H)$ and $\tau_1^*$ maps the $l$-th smallest death value in $D(G)$ to the $l$-th smallest death value in $D(H)$ for all $l$.

*Proof.* Let $G$ and $H$ be networks with $m$ nodes. The topological distance is originally defined as

$$d_{top}(G, H) = \Big( \min_{\tau_0} \sum_{b \in B(G)} \big[b - \tau_0(b)\big]^2 + \min_{\tau_1} \sum_{d \in D(G)} \big[d - \tau_1(d)\big]^2 \Big)^{\frac{1}{2}}.$$

We rewrite the first term as follows.

$$\min_{\tau_0} \sum_{b \in B(G)} \big[b - \tau_0(b)\big]^2 = \min_{\tau_0} \sum_{b \in B(G)} \big[b^2 - 2b\tau_0(b) + \tau_0^2(b)\big]$$

$$= \min_{\tau_0} \sum_{b \in B(G)} -2b\tau_0(b) + C_1$$

$$= \max_{\tau_0} \sum_{b \in B(G)} b\tau_0(b),$$

where $C_1$ is a constant. Note that $\sum_{b \in B(G)} \tau_0^2(b)$ is equivalent to $\sum_{b \in B(H)} b^2$, which is a constant over all bijections $\tau_0$. Similarly for the second term, we have

$$\min_{\tau_1} \sum_{d \in D(G)} \big[d - \tau_1(d)\big]^2 = \max_{\tau_1} \sum_{d \in D(G)} d\tau_1(d).$$

Let $B(G) : x_1 \leq \cdots \leq x_{m-1}$ and $B(H) : y_1 \leq \cdots \leq y_{m-1}$ be the collections of birth values for $G$ and $H$ respectively. By the rearrangement inequality (Mitrinovic, 1970), it is known that

$$x_1 y_1 + \cdots + x_{m-1} y_{m-1} \geq x_{\pi(1)} y_1 + \cdots + x_{\pi(m-1)} y_{(m-1)}$$

for every permutation $x_{\pi(1)}, ..., x_{\pi(m-1)}$ of $x_1, ..., x_{m-1}$. Thus, optimal bijection $\tau_0^*$ maps the $l$-th smallest birth value in $B(G)$ to the $l$-th smallest birth value in $B(H)$. Similarly for the second term, we have $\tau_1^*$ that maps the $l$-th smallest death value in $D(G)$ to the $l$-th smallest death value in $D(H)$. □

### A.2 CLOSED-FORM SOLUTION OF TOPOLOGICAL CENTROID

**Lemma 1.** *Let $G_1, ..., G_n$ be $n$ networks each with $m$ nodes. Let $B(G_i) : b_{i,1} \leq \cdots \leq b_{i,m-1}$ and $D(G_i) : d_{i,1} \leq \cdots \leq d_{i,1+m(m-3)/2}$ be the topology of $G_i$. It follows that the $l$-th smallest birth value of the topological centroid of the $n$ networks is given by the mean of all the $l$-th smallest birth values of such networks, i.e., $\sum_{i=1}^{n} b_{i,l}/n$. Similarly, the $l$-th smallest death value of the topological centroid is given by $\sum_{i=1}^{n} d_{i,l}/n$.*

*Proof.* Let $M$ be a cluster representative. Recall that the topological centroid minimizes the sum of the squared topological distances, i.e.,

$$\min_{M} \sum_{i=1}^{n} d_{top}^2(M, G_i) = \min_{B(M), D(M)} \sum_{i=1}^{n} \Big( \sum_{b \in B(M)} \big[b - \tau_0^*(b)\big]^2 + \sum_{d \in D(M)} \big[d - \tau_1^*(d)\big]^2 \Big).$$

Let $B(M) : \gamma_1 \leq \cdots \leq \gamma_{m-1}$ and $D(M) : \delta_1 \leq \cdots \leq \delta_{1+m(m-3)/2}$ be the topology of $M$. The first term can be expanded as

$$\sum_{i=1}^{n} \Big( \sum_{b \in B(M)} \big[b - \tau_0^*(b)\big]^2 \Big) = \sum_{i=1}^{n} \Big( [\gamma_1 - b_{i,1}]^2 + \cdots + [\gamma_{m-1} - b_{i,m-1}]^2 \Big),$$

which is quadratic and thus can be solved by setting its derivative equal to zero. The solution is given by $\widehat{\gamma}_l = \sum_{i=1}^{n} b_{i,l}/n$. Similarly, the second term can be expanded as

$$\sum_{i=1}^{n} \Big( \sum_{d \in D(M)} \big[d - \tau_1^*(d)\big]^2 \Big) = \sum_{i=1}^{n} \Big( [\delta_1 - d_{i,1}]^2 + \cdots + [\delta_{1+m(m-3)/2} - d_{i,1+m(m-3)/2}]^2 \Big),$$

which is again quadratic. By setting its derivative equal to zero, we find the minimum at $\widehat{\delta}_l = \sum_{i=1}^{n} d_{i,l}/n$.

$\square$

## A.3 TOPOLOGICAL INTERPOLATION

**Lemma 2.**

$$\sum_{h=1}^{k} \min_{\Theta_h} \sum_{G \in C_h} d_{net}(\Theta_h, G) = \sum_{h=1}^{k} \min_{\Theta_h} \Big( (1-\lambda) \sum_{i} \sum_{j>i} \big(\theta_{h,ij} - \overline{w}_{h,ij}\big)^2 + \lambda d_{top}^2\big(\Theta_h, \widehat{M}_{top,h}\big) \Big),$$
(10)

*where $\overline{w}_h = (\overline{w}_{h,ij})$ are edge weights in the sample mean network $\overline{M}_h = (V, \overline{w}_h)$ of cluster $C_h$, and $\widehat{M}_{top,h}$ is the topological centroid of networks in cluster $C_h$.*

*Proof.* Let $G = (V, w) \in C$ and $\Theta = (V, \theta)$. Denote the cardinality of $C$ by $m$. We will show that

$$\min_{\Theta} \sum_{G \in C} d_{net}(\Theta, G) = \min_{\Theta} \Big( (1-\lambda) \sum_{i} \sum_{j>i} \big(\theta_{ij} - \overline{w}_{ij}\big)^2 + \lambda d_{top}^2\big(\Theta, \widehat{M}_{top}\big) \Big).$$

We first rewrite the geometric term as follows.

$$\begin{aligned}
\sum_{G \in C} \sum_{i} \sum_{j>i} \big(\theta_{ij} - w_{ij}\big)^2 &= \sum_{G \in C} \sum_{i} \sum_{j>i} \big(\theta_{ij}^2 - 2\theta_{ij}w_{ij} + w_{ij}^2\big) \\
&= \sum_{i} \sum_{j>i} \Big(m\theta_{ij}^2 - 2\theta_{ij} \sum_{G \in C} w_{ij} + \sum_{G \in C} w_{ij}^2\Big) \\
&= m \sum_{i} \sum_{j>i} \Big(\theta_{ij}^2 - \frac{2\theta_{ij}}{m} \sum_{G \in C} w_{ij}\Big) + C_1 \\
&= m \sum_{i} \sum_{j>i} \Big(\theta_{ij}^2 - \frac{2\theta_{ij}}{m} \sum_{G \in C} w_{ij} + \big(\sum_{G \in C} w_{ij}/m\big)^2\Big) + C_2 \\
&= m \sum_{i} \sum_{j>i} \Big(\theta_{ij} - \sum_{G \in C} w_{ij}/m\Big)^2 + C_2 \\
&= m \sum_{i} \sum_{j>i} \big(\theta_{ij} - \overline{w}_{ij}\big)^2 + C_2,
\end{aligned}$$

where $C_1, C_2$ are constants. Similarly for the topological term, we have

$$\begin{aligned}
\sum_{G \in C} d_{top}^2(\Theta, G) &= \sum_{G \in C} \Big( \sum_{b \in B(\Theta)} \big[b - \tau_0^*(b)\big]^2 + \sum_{d \in D(\Theta)} \big[d - \tau_1^*(d)\big]^2 \Big) \\
&= \sum_{b} \big[mb^2 - 2b \sum_{G \in C} \tau_0^*(b)\big] + \sum_{d} \big[md^2 - 2d \sum_{G \in C} \tau_1^*(d)\big] + C_3 \\
&= m \Big( \sum_{b} \big[b - \sum_{G \in C} \tau_0^*(b)/m\big]^2 + \sum_{d} \big[d - \sum_{G \in C} \tau_1^*(d)/m\big]^2 \Big) + C_4 \\
&= m d_{top}^2(\Theta, \widehat{M}_{top}) + C_4.
\end{aligned}$$

Thus,

$$
\begin{aligned}
\min_{\Theta} \sum_{G \in C} d_{net}(\Theta, G) &= \min_{\Theta}(1-\lambda)\Big(m\sum_i \sum_{j>i} \big(\theta_{ij} - \overline{w}_{ij}\big)^2 + C_2\Big) + \lambda\Big(md_{top}^2(\Theta, \widehat{M}_{top}) + C_4\Big) \\
&= \min_{\Theta} m\Big((1-\lambda)\sum_i \sum_{j>i} \big(\theta_{ij} - \overline{w}_{ij}\big)^2 + \lambda d_{top}^2(\Theta, \widehat{M}_{top})\Big) + C_5 \\
&= \min_{\Theta}\Big((1-\lambda)\sum_i \sum_{j>i} \big(\theta_{ij} - \overline{w}_{ij}\big)^2 + \lambda d_{top}^2(\Theta, \widehat{M}_{top})\Big).
\end{aligned}
$$

$\square$

### A.4 Convergence to a locally optimal partition

**Theorem 1.** *The topological clustering algorithm monotonically decreases $L$ and terminates in a finite number of steps at a locally optimal partition.*

*Proof.* Let $\mathcal{C}^{(t)} = \{C_h^{(t)}\}_{h=1}^k$ be a partition of observed networks with corresponding representatives $\mathcal{M}^{(t)} = \{M_h^{(t)}\}_{h=1}^k$ after the $t$-th iteration. Then,

$$
\begin{aligned}
L(\mathcal{C}^{(t)}, \mathcal{M}^{(t)}) &= \sum_{h=1}^k \sum_{G \in C_h^{(t)}} d_{net}(M_h^{(t)}, G) \\
&\overset{(a)}{\geq} \sum_{h=1}^k \sum_{G \in C_h^{(t+1)}} d_{net}(M_h^{(t)}, G) \\
&\overset{(b)}{\geq} \sum_{h=1}^k \sum_{G \in C_h^{(t+1)}} d_{net}(M_h^{(t+1)}, G) = L(\mathcal{C}^{(t+1)}, \mathcal{M}^{(t+1)}),
\end{aligned}
$$

where (a) follows from the assignment step and (b) follows from applying Lemma 1 (when $\lambda = 1$) and Lemma 2 (when $\lambda \in (0, 1)$ and step size in gradient descent is sufficiently small) to the re-estimation step. Note that $\lambda = 0$ describes conventional edge clustering (MacQueen, 1967). This result shows that the algorithm monotonically decreases $L$. Since the number of distinct partitions is finite and $L$ is monotonically decreasing, eventually $L$ will not be decreased either by (a) the assignment step or (b) re-estimation step and the algorithm terminates. $\square$

## B Experimental Details

### B.1 Supplementary description for candidate methods

#### Methods based on two-dimensional barcodes

The *Wasserstein* method clusters conventional two-dimensional barcodes representing connected components and cycles. To this end, two separate Wasserstein distances are needed to match connected components to connected components and cycles to cycles. The *Wasserstein* method performs clustering using $k$-medoids based on the sum of the two Wasserstein distances for connected components and cycles.

*Bregman Alternating Direction Method of Multipliers* (B-ADMM) (Ye et al., 2017) is a Wasserstein barycenter based algorithm for clustering discrete distributions. We follow the standard approach (Turner et al., 2014) to representing underlying discrete distributions on two-dimensional barcodes in the form of Dirac masses. The high computational complexity of B-ADMM is managed by limiting the clustering to no more than the 50 most persistent cycles.

The two methods require computation of two-dimensional persistence barcodes from networks. We compute a two-dimensional persistence barcode using the approach of Otter et al. (2017) in which edge weights are inverted via the function $f(w) = 1/(1 + w)$. Then a point cloud is obtained from the shortest path distance between nodes. Finally, we compute Rips filtration (Ghrist, 2008) and its corresponding persistence barcode from the point cloud.

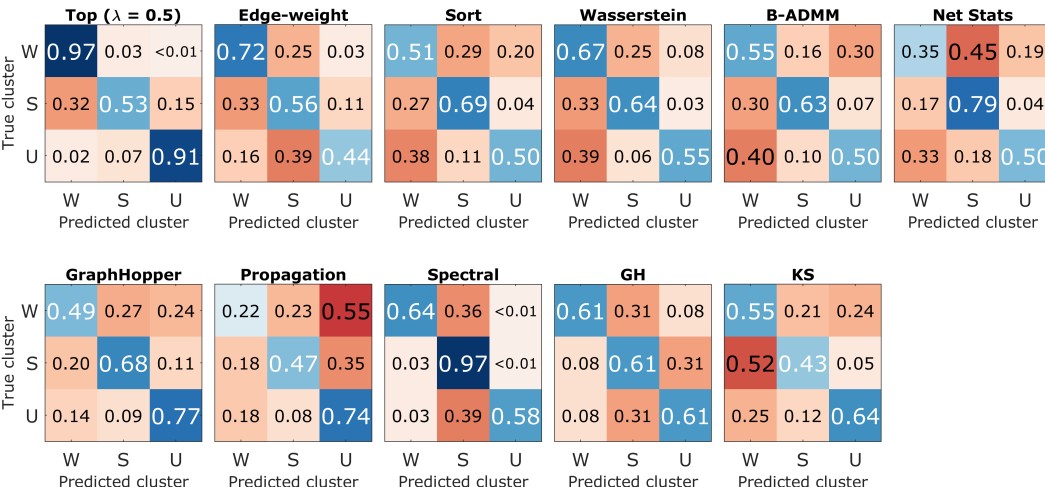

Supplementary Figure 1: Average confusion matrices over 100 random initializations for subject R376. W: wake, S: sedated, U: unresponsive.

METHODS BASED ON GRAPH KERNELS

The two graph kernel methods based on *GraphHopper* and *propagation* kernels require node continuous attributes. We follow the experimental protocol of Borgwardt et al. (2020) in which a node attribute is set to the sum of edge weights incident to the node.

METHODS BASED ON NETWORK DISTANCE MEASURES FOR BRAIN NETWORK ANALYSES

*Gromov-Hausdorff* (GH) distance (Lee et al., 2012) compares dendrogram shape differences in brain networks. GH distance requires brain networks that exhibit metric spaces. To this end, we follow the approach of Otter et al. (2017) for converting a brain network to a metric space. The metric space is obtained by first inverting edge weights via the function $f(w) = 1/(1 + w)$ and then using the shortest path distance between nodes as a metric.

*Kolmogorov-Smirnov* (KS) distance (Chung et al., 2019) compares topological differences in brain networks using their Betti numbers. Two separate KS distances are needed to compare 0-th Betti numbers to 0-th Betti numbers and 1-st Betti numbers to 1-st Betti numbers. The sum of the two KS distances is used to compare brain networks.

We follow the approach of Banks et al. (2020) to comparing brain networks using the operator or *spectral* norm of the difference between network adjacency matrices.

B.2    IMPLEMENTATION OF CANDIDATE METHODS

For all the baseline methods, we used existing implementation codes from authors' publications and publicly available repository websites. We used parameters recommended in the public code without any modification. Code for Wasserstein distance approximation (Lacombe et al., 2018) is available at https://gudhi.inria.fr/. Code for *Bregman Alternating Direction Method of Multipliers* (B-ADMM) method (Ye et al., 2017) is available at https://github.com/bobye/WBC_Matlab. Code for the three network summary statistics used in the *Net Stats* method is available at https://sites.google.com/site/bctnet/. Code for *GraphHopper* and *propagation* graph kernels is available at https://github.com/ysig/GraKeL.

B.3    SUPPLEMENTARY RESULTS FOR SIMULATED AND MEASURED DATASETS

For simulated networks, clustering performance comparison is reported in Supplementary Table 1. For measured brain networks, confusion matrices of all the candidate methods for subject R376 (see Supplementary Table 2) are reported in Supplementary Figure 1.

Supplementary Table 1: Clustering performance comparison for simulated networks with $|V| = 60$ nodes. (a) average accuracy and (b) average $p$-values for various parameter settings of $m$ (number of modules) and $r$ (within-module connection probability).

| $m$ | $r$ | Top ($\lambda = 1$) | Edge-weight | Sort | Wasserstein | B-ADMM | Net Stats | GraphHopper | Propagation |
|---|---|---|---|---|---|---|---|---|---|
| 2/3/5 | 0.6 | $0.75 \pm 0.06$ | $0.46 \pm 0.04$ | $0.76 \pm 0.06$ | $0.57 \pm 0.06$ | $0.61 \pm 0.05$ | $0.75 \pm 0.06$ | $0.69 \pm 0.08$ | $0.60 \pm 0.12$ |
| | 0.7 | $0.95 \pm 0.06$ | $0.62 \pm 0.13$ | $0.94 \pm 0.08$ | $0.77 \pm 0.08$ | $0.81 \pm 0.08$ | $0.96 \pm 0.02$ | $0.90 \pm 0.12$ | $0.75 \pm 0.13$ |
| | 0.8 | $0.97 \pm 0.10$ | $0.75 \pm 0.17$ | $0.97 \pm 0.10$ | $0.88 \pm 0.12$ | $0.92 \pm 0.08$ | $0.98 \pm 0.09$ | $0.98 \pm 0.09$ | $0.80 \pm 0.16$ |
| | 0.9 | $0.94 \pm 0.13$ | $0.81 \pm 0.17$ | $0.93 \pm 0.14$ | $0.90 \pm 0.15$ | $0.96 \pm 0.09$ | $0.95 \pm 0.12$ | $0.97 \pm 0.13$ | $0.82 \pm 0.20$ |
| 2/5/10 | 0.6 | $0.78 \pm 0.07$ | $0.44 \pm 0.05$ | $0.78 \pm 0.07$ | $0.63 \pm 0.06$ | $0.66 \pm 0.05$ | $0.77 \pm 0.07$ | $0.73 \pm 0.11$ | $0.65 \pm 0.14$ |
| | 0.7 | $0.85 \pm 0.13$ | $0.59 \pm 0.10$ | $0.85 \pm 0.12$ | $0.81 \pm 0.09$ | $0.86 \pm 0.07$ | $0.89 \pm 0.10$ | $0.85 \pm 0.21$ | $0.81 \pm 0.12$ |
| | 0.8 | $0.90 \pm 0.14$ | $0.72 \pm 0.13$ | $0.89 \pm 0.15$ | $0.88 \pm 0.14$ | $0.91 \pm 0.13$ | $0.91 \pm 0.14$ | $0.85 \pm 0.26$ | $0.85 \pm 0.15$ |
| | 0.9 | $0.92 \pm 0.14$ | $0.78 \pm 0.15$ | $0.92 \pm 0.14$ | $0.92 \pm 0.14$ | $0.91 \pm 0.15$ | $0.90 \pm 0.15$ | $0.73 \pm 0.33$ | $0.85 \pm 0.17$ |

(a) Average accuracy

| $m$ | $r$ | Top ($\lambda = 1$) | Edge-weight | Sort | Wasserstein | B-ADMM | Net Stats | GraphHopper | Propagation |
|---|---|---|---|---|---|---|---|---|---|
| 2/3/5 | 0.6 | $< 0.001$ | $0.28 \pm 0.26$ | $< 0.001$ | $0.02 \pm 0.12$ | $0.003 \pm 0.021$ | $< 0.001$ | $0.02 \pm 0.14$ | $0.09 \pm 0.27$ |
| | 0.7 | $< 0.001$ | $0.07 \pm 0.21$ | $< 0.001$ | $< 0.001$ | $< 0.001$ | $< 0.001$ | $0.04 \pm 0.20$ | $0.01 \pm 0.10$ |
| | 0.8 | $< 0.001$ | $0.03 \pm 0.13$ | $< 0.001$ | $< 0.001$ | $< 0.001$ | $< 0.001$ | $0.02 \pm 0.14$ | $0.04 \pm 0.20$ |
| | 0.9 | $< 0.001$ | $0.01 \pm 0.09$ | $< 0.001$ | $< 0.001$ | $< 0.001$ | $< 0.001$ | $0.04 \pm 0.20$ | $0.08 \pm 0.27$ |
| 2/5/10 | 0.6 | $< 0.001$ | $0.37 \pm 0.33$ | $< 0.001$ | $0.004 \pm 0.034$ | $< 0.001$ | $< 0.001$ | $0.06 \pm 0.24$ | $0.12 \pm 0.31$ |
| | 0.7 | $< 0.001$ | $0.05 \pm 0.14$ | $< 0.001$ | $< 0.001$ | $< 0.001$ | $< 0.001$ | $0.14 \pm 0.35$ | $0.01 \pm 0.10$ |
| | 0.8 | $< 0.001$ | $0.03 \pm 0.15$ | $< 0.001$ | $< 0.001$ | $< 0.001$ | $< 0.001$ | $0.20 \pm 0.40$ | $0.03 \pm 0.17$ |
| | 0.9 | $< 0.001$ | $0.004 \pm 0.039$ | $< 0.001$ | $< 0.001$ | $< 0.001$ | $< 0.001$ | $0.41 \pm 0.49$ | $0.04 \pm 0.20$ |

(b) Average $p$-values

Supplementary Table 2

| Subject | Age | Gender | Network size |
|---|---|---|---|
| R369 | 30 | M | 199 |
| L372 | 34 | M | 174 |
| R376 | 48 | F | 189 |
| B384 | 38 | M | 89 |
| R399 | 22 | F | 175 |
| L400 | 59 | F | 126 |
| L403 | 56 | F | 194 |
| L405 | 19 | M | 127 |
| L409 | 31 | F | 160 |
| L423 | 51 | M | 152 |
| L514 | 46 | M | 118 |

## C  BRAIN NETWORK DATASET

Brain network data were obtained from eleven neurosurgical patients (5 female, ages 19 - 59 years old, median age 38 years old; Supplementary Table 2). The patients had been diagnosed with medically refractory epilepsy and were undergoing chronic invasive intracranial electroencephalography (iEEG) monitoring to identify potentially resectable seizure foci. All human subjects experiments were carried out in accordance with The Code of Ethics of the World Medical Association (Declaration of Helsinki) for experiments involving humans. The research protocols were approved by the University of Iowa Institutional Review Board and the National Institutes of Health. Written informed consent was obtained from all subjects. Research participation did not interfere with acquisition of clinically required data. Subjects could rescind consent at any time without interrupting their clinical evaluation. Recordings were made using subdural and depth electrodes (Ad-Tech Medical, Oak Creek, WI) placed solely on the basis of clinical requirements, as determined by the team of epileptologists and neurosurgeons. The brain network dataset was based on recordings made in the operating room prior to electrode removal, before and during induction of general anesthesia with propofol. Details of the experimental procedure, recording, and electrophysiological data analysis are available in (Banks et al., 2020).

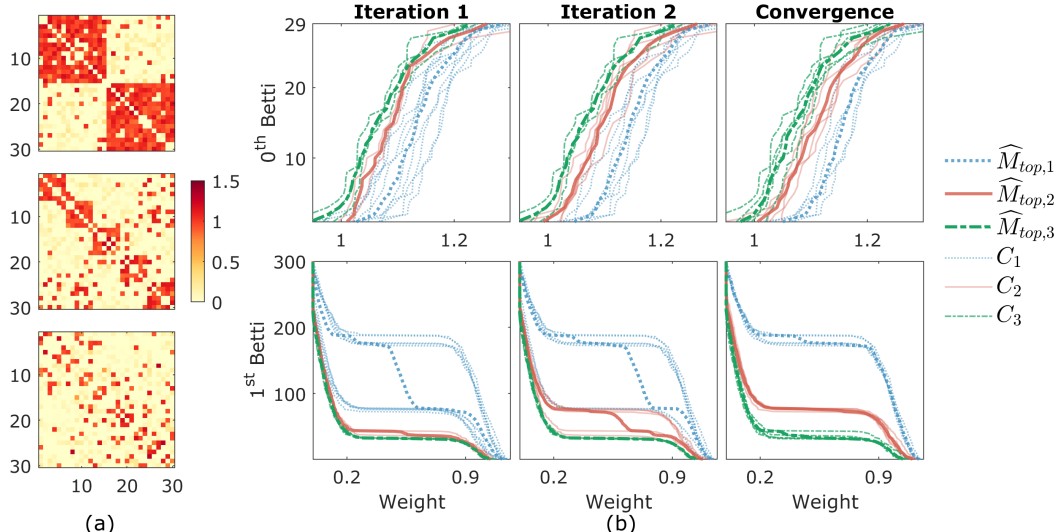

Supplementary Figure 2: Toy illustration of performing topological clustering using only topological centroids ($\lambda = 1$). Clustering is performed on 15 networks: five each with $m = 2, 5$ and 10 modules. All networks use $|V| = 30$, within module connection probability $r = 0.9$ and edge weight standard deviation $\sigma = 0.1$. (a) Network examples for $m = 2$ (top), 5 (middle) and 10 (bottom) modules. (b) Illustration of topological clustering. Topological centroids $\widehat{M}_{top,i}$ (thick lines) and individual network topologies are visualized using Betti numbers as a function of edge weights (filtration values). The algorithm converges in three iterations and the final partition (last column) perfectly matches the ground truth, i.e., $C_1, C_2$ and $C_3$ corresponds to $m = 2, 5$ and 10.

## D   TOY ILLUSTRATION OF TOPOLOGICAL CLUSTERING

Supplementary Figure 2 illustrates a cluster analysis of the proposed topological approach using only topological centroids ($\lambda = 1$) on a toy dataset generated using random modular networks described in Section 4 in the main body of the paper.

