# OpenReview forum: "Fast topological clustering with Wasserstein distance"
_ICLR.cc/2022/Conference — ICLR 2022 Poster_

### Official Review · Reviewer_FgNK · 2021-10-19

**Correctness:** 4
**Technical Novelty And Significance:** 2
**Empirical Novelty And Significance:** 2
**Recommendation:** 8
**Confidence:** 5

**Main Review:**

The presentation of the paper is clear and the paper is well-written. The problem of clustering persistence diagrams (PDs) has attracted some attention over the recent years and is not a trivial one. To overcome the main computational challenges that arise when having to compute distances/barycenters of PDs, authors propose to consider simpler persistence diagrams that are one-dimensional (that is all points in the PDs have coordinates of the form $(-\infty,d_i)$ or $(b_i,+\infty)$). It is well-known that in one dimension, optimal transport essentially boils down to sorting algorithms, so that all the computational tasks (computing distances/barycenters) become trivial. A naive implementation of k-means is then proposed using the distance between PDs, that is able to attain a local minimum of the energy functional at stake.

My main issue with the article is that they never clearly acknowledge that they work with "simple" PDs that have a very particular form, and that this makes the problem much easier to solve. Those simpler PDs appear because the filtration F that they build on top of the weighted graph does not contain any two-dimensional simplices: as such the points of the PDs PD(F) are of the form $(-\infty,d_i)$. What is usually done is to consider a filtration G with two-dimensional simplices appearing and "filling" the loops of the graph at some point, the corresponding PD PD(G) having points of the form $(b_i,d_i)$. To put it another way, we are considering a PD, PD(F), obtained by considering the projection of the points of PD(G) on the y-axis. We are therefore losing information while doing this. I have nothing against considering a simpler object if this allows one to also simplify computations, but this needs to appear explicitly in the paper.

The question is then: "What exactly do we lose by considering the simple filtration F instead of G?". I do not expect a mathematical treatment here, but it would seem fair to compare experimentally the method proposed in the paper with the alternative method with the filtration G used. More precisely, in the case $\lambda =1$ (considering the topological distance), one could cluster graphs applying the k-mean algorithm on the PDs built with G, computing barycenters of PDs using one of the standard approaches [1,2]. We expect the second method to take more time, but to be more precise. Of course, a good selling point for the paper would be to show that using the filtration G instead of the filtration F (i) takes much more time and (ii) does not make us gain that much accuracy. Such an empirical confirmation should really be at the core of the paper in my opinion.

Overall, the idea of considering simpler objects on which computations can be done is appealing, but this needs to be backed up by either theoretical or empirical evidence. With more experiments, I would be willing to improve my grade.


MINOR COMMENTS:
- p.1 "Wasserstein distance can be used to interpolate networks while preserving topological structure," please give a ref
- Eq (4): this would rather be d_net^2, if we want d_net to be a distance.
- p.4 How is this algorithm different from Lloyd's algorithm (in a general metric space)?
- p.5 bottom: comparing to k-medoids using the W2 distance instead of the bottleneck distance appears to be fairer, as it will be more able to assess the fine structure in the geometry of the points of the PDs.
- Fig 5: this behavior is interesting, and should be commented on: taking $\lambda = 0$ is a bad idea, but then, any positive value (that is still not too close to 1) appears to give a similar score. Are there any explanations about why this is the case?
- On the experiments: it seems weird that they do not exist better alternative methods that are **at least** able to obtain a score similar to the one obtained by the paper's method.


[1] Geometry Helps to Compare Persistence Diagrams - Michael Kerber, Dmitriy Morozov, Arnur Nigmetov

[2] Large Scale computation of Means and Clusters for Persistence Diagrams using Optimal Transport - Theo Lacombe, Marco Cuturi, Steve OUDOT

**Summary Of The Paper:**

Given a weighted graph, authors construct a filtration on top of it that is associated with a persistence diagram. This persistence diagram summarizes the topology of the weighted graph. Authors propose then to cluster different graphs using either the distance between the persistence diagrams (that they call the topological distance) or an interpolation between the L2 distance between the weights and the topological distance.

The relevance of this algorithm is then showcased on synthetic examples and real-world datasets.


**Summary Of The Review:**

The proposed method to cluster graphs using persistence diagrams is computationally efficient, but at the price of losing some topological information in the process. This can definitely be a good approach in some problems, but I would like to have some empirical evidence that "we do not lose too much information" in the process, that is that this procedure competes with the (slower) one where we take into account the complete topological information.

---

> ### Author Response · Authors · 2021-11-23
> **Response to Reviewer FgNK [1/3]**
>
> We thank the reviewer for the detailed, thoughtful, and constructive review. The reviewer’s suggestions are consistent with the other reviews. We have addressed each recommendation as described below and are confident that the revisions have significantly improved the manuscript.
>
> The reviewers’ comments are numbered and summarized in _italics_ below. The response is in ordinary font.
>
> _1. My main issue with the article is that they never clearly acknowledge that they work with "simple" PDs that have a very particular form, and that this makes the problem much easier to solve. Those simpler PDs appear because the filtration F that they build on top of the weighted graph does not contain any two-dimensional simplices: as such the points of the PDs PD(F) are of the form (−∞,di). What is usually done is to consider a filtration G with two-dimensional simplices appearing and "filling" the loops of the graph at some point, the corresponding PD PD(G) having points of the form (bi,di). To put it another way, we are considering a PD, PD(F), obtained by considering the projection of the points of PD(G) on the y-axis. We are therefore losing information while doing this. I have nothing against considering a simpler object if this allows one to also simplify computations, but this needs to appear explicitly in the paper._
>
> The reviewer is correct that the PDs have a simple form because of the nature of the filtration and limitation to connected components and cycles. The revision more clearly acknowledges these points which were also raised by the other reviewers.
>
> Specifically:
> 1. The limitation to considering 1-skeletons are noted in the middle of Section 2.1.
> 2. The possibility of other filtrations are noted at the end of Section 2.1.
> 3. The second paragraph of Section 2.2 begins by stating that the discussion to follow is for an edge-weight threshold graph filtration as defined in Section 2.1.
> 4. The last sentence of the second paragraph of Section 2.2 notes that other filtrations do not share the monotonicity property that results in one-dimensional barcode representations.
> 5. Section 3.1 is retitled "Topological Distance Simplification" and begins by noting that the simplification of the 2-Wasserstein distance is a consequence of the assumptions.

---

> > ### Author Response · Authors · 2021-11-23
> > **Response to Reviewer FgNK [2/3]**
> >
> > _2. The question is then: "What exactly do we lose by considering the simple filtration F instead of G?". I do not expect a mathematical treatment here, but it would seem fair to compare experimentally the method proposed in the paper with the alternative method with the filtration G used. More precisely, in the case λ=1 (considering the topological distance), one could cluster graphs applying the k-mean algorithm on the PDs built with G, computing barycenters of PDs using one of the standard approaches [1,2]. We expect the second method to take more time, but to be more precise. Of course, a good selling point for the paper would be to show that using the filtration G instead of the filtration F (i) takes much more time and (ii) does not make us gain that much accuracy. Such an empirical confirmation should really be at the core of the paper in my opinion._
> >
> > The revised manuscript includes several new baseline methods to provide empirical confirmation of the effectiveness of the proposed method, including two methods based on two-dimensional barcodes resulting from the well-known Rips filtration. One of these methods is k-medoids clustering based on the sum of Wasserstein distances for connected components and cycles.  The other is based on a previously proposed Wasserstein barycenter method.  Two approaches based on graph kernels and a method based on graph theoretic measures have also been added.
> >
> > The computational costs of methods based on two-dimensional barcodes are indeed very significant, and prevented us from including results for the 120-node simulated networks.  There was not sufficient time in the revision period to run these cases with our computing system.  Reporting computational complexity using run time is difficult to do in a meaningful, enduring manner, so we chose to refer to the literature describing the computational costs for the Rips filtration (Otter et al., 2017, A roadmap for the computation of persistent homology), Wasserstein distance approximation (Lacombe et al., 2018, Large scale computation of means and clusters for persistence diagrams using optimal transport) and Wasserstein Barycenter method (Ye et al., 2017, Fast Discrete Distribution Clustering Using Wasserstein Barycenter with Sparse Support).
> >
> > The results indeed show that the proposed approach does not result in loss of accuracy while being much, much faster in computation.  The methods based on two-dimensional barcodes involve certain approximations in order to be computationally tractable, and these approximations reduce the sensitivity of the methods to subtle topological details.  Hence, the performance degrades as the modularity strength (determined by r) in the simulated networks decreases.  In contrast, the simple filtration we propose appears to be relatively robust, even though it is based on a limited set of topological features.
> >
> > _3. Overall, the idea of considering simpler objects on which computations can be done is appealing, but this needs to be backed up by either theoretical or empirical evidence. With more experiments, I would be willing to improve my grade._
> >
> > Thank you for the suggestions.  We trust the additional experiments and clarification of assumptions satisfactorily address your request.

---

> > > ### Author Response · Authors · 2021-11-23
> > > **Response to Reviewer FgNK [3/3]**
> > >
> > > _4. MINOR COMMENTS:_
> > > * _p.1 "Wasserstein distance can be used to interpolate networks while preserving topological structure," please give a ref_
> > >
> > > This property was shown in (Songdechakraiwut et al., 2021, Topological learning and its application to multimodal brain network integration) and the citation is in the revised manuscript.
> > >
> > > * _Eq (4): this would rather be d_net^2, if we want d_net to be a distance._
> > >
> > > We have adopted the reviewer’s preference for d_{net} to be a distance and defined d^2_{net} in Eq. (4).
> > >
> > > * _p.4 How is this algorithm different from Lloyd's algorithm (in a general metric space)?_
> > >
> > > The distinction lies in how the two methods select cluster centroids. k-means selects a cluster sample mean as a centroid, which does not necessarily represent a meaningful network, while the proposed method constrains the search of cluster centroids to a space of meaningful networks through topological interpolation (Lemma 2). This distinction is now explicitly pointed out in the paragraph following Lemma 2.
> > >
> > > * _p.5 bottom: comparing to k-medoids using the W2 distance instead of the bottleneck distance appears to be fairer, as it will be more able to assess the fine structure in the geometry of the points of the PDs._
> > >
> > > Recommendation adopted, as noted in our response to comment 2.
> > >
> > > * _Fig 5: this behavior is interesting, and should be commented on: taking λ=0 is a bad idea, but then, any positive value (that is still not too close to 1) appears to give a similar score. Are there any explanations about why this is the case?_
> > >
> > > The new Figure 6 sheds additional light on this behavior, as well as the added discussion of the role of lambda.  Observe that most subjects show improved performance for lambda > 0, and then the performance begins to degrade as lambda approaches 1.  This suggests that both topology and geometry are helpful in clustering these types of brain networks and ignoring either results in reduced performance.  This is reasonable given the fact that many brain areas are functionally specialized - we expect that topological changes associated with changes in consciousness will be different in distinct brain regions.
> > >
> > > * _On the experiments: it seems weird that there do not exist better alternative methods that are at least able to obtain a score similar to the one obtained by the paper's method._
> > >
> > >
> > > We revised the simulation study to examine performance as a function of the strength of the modularity, controlled by r,  in the simulated networks. The new Figure 2 illustrates that network structure becomes increasingly subtle as r decreases.  Many methods perform very well when there is pronounced modularity.  The performance advantages of the proposed method occur with more nuanced network structure (r = 0.7, 0.6).
> > >
> > > _[1] Geometry Helps to Compare Persistence Diagrams - Michael Kerber, Dmitriy Morozov, Arnur Nigmetov_
> > >
> > > _[2] Large Scale computation of Means and Clusters for Persistence Diagrams using Optimal Transport - Theo Lacombe, Marco Cuturi, Steve OUDOT_

---

> > > > ### Comment · Reviewer_FgNK · 2021-11-29
> > > > **Response to authors**
> > > >
> > > > The new version addresses all my previous concerns. I improved my score accordingly.
> > > >
> > > > Minor remarks:
> > > > - "The topological patterns exhibited by many real-world networks motivate THE development of topology-based methods for assessing the similarity of networks."
> > > > - The filtration you consider in section 4 is not the Rips filtration. Usually, the Rips filtration is built on top of a point cloud, with the matrix (w_ij) being given by the distances between the points. This filtration could for instance be called the "clique complex filtration" or the "clique filtration" (this terminology appears to have already been used in previous papers).
> > > > - I find it very surprising that the Wasserstein method performs worse than the Top method. Do you think this is due to the fact that only the approximate distances are computed with the former? Intuitively, considering persistence diagrams with more information should improve the accuracy. I would be happy to have any feedback from the authors on this.

---

> > > > > ### Author Response · Authors · 2021-12-01
> > > > > **Response to Reviewer FgNK**
> > > > >
> > > > > _The new version addresses all my previous concerns. I improved my score accordingly._
> > > > >
> > > > > We thank the reviewer for improving the score.
> > > > >
> > > > > _Minor remarks:_
> > > > >
> > > > > _"The topological patterns exhibited by many real-world networks motivate THE development of topology-based methods for assessing the similarity of networks."_
> > > > >
> > > > > We have corrected the missing article as suggested.
> > > > >
> > > > > _The filtration you consider in section 4 is not the Rips filtration. Usually, the Rips filtration is built on top of a point cloud, with the matrix (w_ij) being given by the distances between the points. This filtration could for instance be called the "clique complex filtration" or the "clique filtration" (this terminology appears to have already been used in previous papers)._
> > > > >
> > > > > We agree with the reviewer that clique complex and filtration are more appropriate terms for networks and thus use the terms as suggested.
> > > > >
> > > > > _I find it very surprising that the Wasserstein method performs worse than the Top method. Do you think this is due to the fact that only the approximate distances are computed with the former? Intuitively, considering persistence diagrams with more information should improve the accuracy. I would be happy to have any feedback from the authors on this._
> > > > >
> > > > > We agree with the reviewer that the nature of the approximation used to estimate the Wasserstein distance is one likely cause for the performance loss. This hypothesis could explain the high standard deviation of the Wasserstein method for the cases of pronounced modularity (r = 0.9, 0.8). The approximation algorithm is iterative and variations in convergence could explain the variability in performance. For the cases of more nuanced modularity (r = 0.7, 0.6), we also observe that less sophisticated methods such as sort, Net Stats, and edge histogram methods perform better than the Wasserstein method. This result suggests that simpler constructs are more effective in noisy settings. We conjecture that the approximate distances in two-dimensional persistence diagrams are not sensitive to the more subtle topological features of these noisy cases. In other words, the combination of noise and approximation error leads to degraded performance. While these seem like reasonable hypotheses, considerable additional work is needed to obtain compelling evidence for the performance implications of approximating Wasserstein distance for two-dimensional persistence diagrams.

---

### Official Review · Reviewer_NiBH · 2021-10-30

**Correctness:** 3
**Technical Novelty And Significance:** 3
**Empirical Novelty And Significance:** 3
**Recommendation:** 8
**Confidence:** 5

**Main Review:**

This is a paper about a timely and highly relevant topic, and I find
myself to be very excited about its results and promises. The paper
stands on firm theoretical footing: the proposed algorithm is described
in sufficient detail to be reproducible and a brief analysis of the
theory behind it is provided as well. The current write-up is already
very clear, and only suffers from minor formulation issues or clarity
aspects, on which I shall subsequently comment. The primary weakness of
the paper lies in the experimental section; here, I would suggest
improvements in terms of the discussion here, because the jump between
simulated networks and real-world networks might necessitate some
experiment 'in-between.'

## Detailed Comments

- When introducing graph filtrations, consider discussing that such
  filtrations can also be generated using specific descriptor functions,
  such as a heat kernel (Carrière et al., [PersLay: A Neural Network
  Layer for Persistence Diagrams and New Graph Topological
  Signatures](http://proceedings.mlr.press/v108/carriere20a/carriere20a.pdf)),
  or they can even be *learned* in a task-specific manner (Hofer et al.,
  [Graph Filtration
  Learning](https://proceedings.mlr.press/v119/hofer20b.html)).
  Moreover, it could e discussed that the presented filtration is only
  one way of filtering over the network; this could be managed by
  a brief discussion on sublevel or superlevel set filtrations, for
  instance.

- Moreover, it should be made clear that, unless I am mistaken, the type
  of filtration described in the paper is a **highly specific** type of
  filtration in the sense that not all filtrations will result in the
  respective creation--destruction tuples (or birth--death tuples,
  respectively). For instance, if a descriptor function (like the
  aforementioned heat kernel) is used, creation tuples might be of the
  generic form $(a, b)$ with $a, b \in \mathbb{R}$, as opposed to one of
  them being infinite. I am mentioning this point not out of pedantry,
  but because it should be clarified that the simplifications of the
  Wasserstein distance later on are *contingent* on the tuples being of
  that form. To some extent, this limits the general applicability of
  the approach: if higher-dimensional simplices are present in the
  network, cycles could be assigned finite persistence values (the same
  holds for the calculation of extended persistence, or the use of
  another descriptor function that does not necessarily result in
  a fully-connected graph). It would immensely strengthen the current
  write-up in the paper if these special properties were at least
  briefly discussed.

- To add to this, such special properties could also potentially be
  mentioned in Section 3.1 when discussing how to derive the distance
  function.

- The notation for solving the optimisation problem could also be
  improved. At first, the networks are described with a weight function
  $\mathbf{w}$ and $\mathbf{u}$, respectively. Later on, this notation
  is changed to include $\mathbf{\theta}$ for a cluster representative.
  This should be clarified; I understand that $\mathbf{\theta}$ is just
  another weight function, is this correct?

- How should the regularisation strength $\lambda$ be chosen in
  practice? I really like the discussion of the 'extreme cases' that one
  can obtain, but a more specific discussion later on in the
  experimental section would be appreciated. I would also suggest to
  consider the inclusion of a 'target function' plot in which $\lambda$
  is being varied; it would be interesting to see how the overall
  accuracy, for instance, depends on this parameter. Knowledge about the
  stability of a specific choice for $\lambda$ would help to instil more
  trust into the method.

- As for the experimental section, I think it would strengthen the paper
  if more details were provided, in particular when it comes to the
  validation experiment. It is astonishing to me that existing methods
  all *fail* to cluster such networks with pronounced modularities.
  I would suggest to discuss this.

- I also do not understand how the comparison partner methods are
  working here; are all weights of the network being vectorised into
  a feature vector of the same size? This points needs elucidation
  because although the number of vertices is always the same, such
  a vectorisation does not account for the fact that a graph-level
  representation needs to be permutation-invariant here; node $1$ in one
  network is *not* the same as node $1$ in another network. If the
  vectorisation strategy does *not* account for this, it might be an
  explanation for the bad performance of other methods.

- Moreover, when it comes to the comparison of different methods, simple
  statistical baselines (for instance a feature vector based on degrees,
  overall modularity of nodes, clustering coefficient, etc.) could also
  be employed (if they are unsuitable, this should at least be
  discussed). I would anticipate that $k$-means based on network summary
  statistics might perform better than $k$-means using vectorised
  weights? It would also be interesting to see a simple graph kernel
  being employed here, for instance a graph kernel using a graph-level
  histogram; since the networks are not excessively large, adding such
  additional comparison partners should be still be possible.

- The same comments apply, *mutatis mutandis*, to the discussion of the
  brain network data set. Here, it would be imperative to show different
  results for different values of $\lambda$. Moreover, given the large
  standard deviations, it seems that almost all methods are performing
  similarly here; is there a reason for such a high variance in the
  results? This should be briefly discussed.

- Out of curiosity: would such an experiment benefit from a more fuzzy
  definition of clusters? The method should potentially allow for this,
  I think, and this might result in better clusters, in particular when
  it comes to an unclear delineation between wake and sedated states.

## Typos, word-smithing, etc.

Overall, the writing in this paper flows well and the language is clear.
There are some cases in which articles are missing or a plural form
might be employed, but this does not prevent a clear understanding of
the text. Here are two examples of this (additional ones can be found
throughout the text; since some of them boil down to personal
preference, I will refrain from listing them):

- 'cycle structure is ubiquitous' --> 'cycles are ubiquitous'

- 'takes infinitely large amount of work' --> 'takes an infinitely large amount'



**Summary Of The Paper:**

This paper presents a novel algorithm for clustering networks based on
their topological properties. To this end, an algorithm based on
persistent homology (a method from computational topology that permits
the calculation of multi-scale features of unstructured and structured
data sets) is introduced. The key feature of the algorithm is that it
manages to substantially reduce the cost of 'matching' topological
features between two different data sets, thus making their comparison
and similarity assessment computationally feasible. The paper provides
an algorithm that can employ *both* geometrical and topological features
of a network (with an appropriate regularisation term); experiments
demonstrate the general utility of the algorithm.


**Summary Of The Review:**

Overall, this is a very interesting paper with a very relevant and
timely topic (potentially high impact). The theory is strong and
well-explained, even though the current write-up suffers from minor
formulation issues, which, however, do not detract from the overall
clarity of the message. I lean towards accepting the paper with the
proviso that clarity is improved and, most importantly, my comments
concerning the experiments are being considered. At present, it is
slightly unclear for non-expert readers to judge the overall utility of
the algorithm since the choice of comparison partners and their training
have yet to be clarified.

**Update after rebutall**: I thank the authors for their updates to the paper;
my concerns were addressed, and I am happy to raise my score.

---

> ### Author Response · Authors · 2021-11-23
> **Response to Reviewer NiBH [1/3]**
>
> We thank the reviewer for the detailed, thoughtful, and constructive review. The reviewer’s suggestions are consistent with the other reviews. We have addressed each recommendation as described below and are confident that the revisions have significantly improved the manuscript.
>
> The reviewers’ comments are numbered and summarized in _italics_ below. The response is in ordinary font.
>
> _1. When introducing graph filtrations, consider discussing that such filtrations can also be generated using specific descriptor functions, such as a heat kernel (Carrière et al., PersLay: A Neural Network Layer for Persistence Diagrams and New Graph Topological Signatures), or they can even be learned in a task-specific manner (Hofer et al., Graph Filtration Learning). Moreover, it could be discussed that the presented filtration is only one way of filtering over the network; this could be managed by a brief discussion on sublevel or superlevel set filtrations, for instance._
>
> Citations to these more general forms of graph filtrations have been added to the end of Section 2.1.
>
> _2. Moreover, it should be made clear that, unless I am mistaken, the type of filtration described in the paper is a highly specific type of filtration in the sense that not all filtrations will result in the respective creation--destruction tuples (or birth--death tuples, respectively). For instance, if a descriptor function (like the aforementioned heat kernel) is used, creation tuples might be of the generic form (a,b) with a,b∈R, as opposed to one of them being infinite.  I am mentioning this point not out of pedantry, but because it should be clarified that the simplifications of the Wasserstein distance later on are contingent on the tuples being of that form. To some extent, this limits the general applicability of the approach: if higher-dimensional simplices are present in the network, cycles could be assigned finite persistence values (the same holds for the calculation of extended persistence, or the use of another descriptor function that does not necessarily result in a fully-connected graph). It would immensely strengthen the current write-up in the paper if these special properties were at least briefly discussed._
>
> The second paragraph of Section 2.2 is modified to limit the observations about birth and death of connected components and cycles to the specific type of filtration assumed in the paper. We explicitly note at the end of the second paragraph that the one-dimensional barcode used in the paper does not necessarily apply to other types of graph filtrations.
>
> _3. To add to this, such special properties could also potentially be mentioned in Section 3.1 when discussing how to derive the distance function._
>
> Section 3.1 has been retitled and the initial sentence modified to note the specific assumptions associated with simplification of the Wasserstein distance.
>
> _4. The notation for solving the optimization problem could also be improved. At first, the networks are described with a weight function w and u, respectively. Later on, this notation is changed to include θ for a cluster representative. This should be clarified; I understand that θ is just another weight function, is this correct?_
>
> We have followed the reviewer’s suggestion to simplify the notation and have used u for the cluster representative weights when deriving the gradient.
>
> _5. How should the regularisation strength λ be chosen in practice? I really like the discussion of the 'extreme cases' that one can obtain, but a more specific discussion later on in the experimental section would be appreciated. I would also suggest considering the inclusion of a 'target function' plot in which λ is being varied; it would be interesting to see how the overall accuracy, for instance, depends on this parameter. Knowledge about the stability of a specific choice for λ would help to instil more trust into the method._
>
> The revised manuscript includes new graphs showing performance of the proposed method as a function of lambda.  The revised Figure 4 represents the extreme cases of simulated networks - very strong modular structure and very subtle modular structure.  The revised Figure 6 shows average performance on the individual subject level.  Discussion of the dependence of performance on lambda has been added to the manuscript.  While the simulated and brain network clustering problems perform best with different ranges of lambda, neither one of them is very sensitive to the exact value chosen.

---

> > ### Author Response · Authors · 2021-11-23
> > **Response to Reviewer NiBH [2/3]**
> >
> > _6. As for the experimental section, I think it would strengthen the paper if more details were provided, in particular when it comes to the validation experiment. It is astonishing to me that existing methods all fail to cluster such networks with pronounced modularities. I would suggest discussing this._
> >
> > The reviewer’s conclusion that the modularities were pronounced in the simulation study are likely due to assuming that the simulation study was based on the r = 0.9 probability of within module connection networks illustrated in the original Figure 2, the toy illustration of clustering using topology.  The simulations actually used r = 0.6, a very nuanced level of modularity.  This explains the poor performance of the methods considered.
> >
> > We have addressed this potential misunderstanding by augmenting the simulations to explicitly show the dependence of performance on the effective “topology to noise” ratio.  This ratio is changed by varying “r”, the within module connection probability, to include r = 0.9 (very pronounced modularities), r = 0.8, r = 0.7, and r = 0.6 (very subtle modularities). Example networks corresponding to these varying modularity strengths are shown in the revised Figure 2. We also moved the toy illustration of clustering using topology to the supplementary material.
> >
> > Note that all the methods perform relatively well with pronounced modularity (r = 0.9), but as the modularity strength or topology to noise ratio decreases with decreasing r, clustering performance also decreases.  The proposed topological clustering algorithm (and the sorted Euclidean distance based clustering) show significant improvement in clustering accuracy for the more nuanced modularity associated with r = 0.7 and r = 0.6.
> >
> > _7. I also do not understand how the comparison partner methods are working here; are all weights of the network being vectorised into a feature vector of the same size? This points needs elucidation because although the number of vertices is always the same, such a vectorisation does not account for the fact that a graph-level representation needs to be permutation-invariant here; node 1 in one network is not the same as node 1 in another network. If the vectorisation strategy does not account for this, it might be an explanation for the bad performance of other methods._
> >
> > We have included a different set of comparison methods in the revision and have attempted to describe them concisely in the manuscript and provide their implementation details in the supplementary material.  In particular, spectral clustering has been removed as it uses a vectorial representation of edge weights.
> >
> > The additional results and experiments included in the revision shed some light on the comparison partner method performance. As noted in the response to comment 6, most comparison methods perform very well with very strong modularity structure and then the performance degrades as the modularity structure becomes more nuanced.  This shows that the methods are implemented in a manner that allows them to be successful, though the proposed method’s performance is somewhat more robust to decreasing strength of modularity.
> >
> > Note that because some of these additional methods are extremely computational intensive and we added simulations showing performance as a function of the (effective) topology to noise level, there was not sufficient time in the revision period to obtain results with the new methods for the larger simulated network (120 nodes) provided in the original submission.  However, the simulation study with 60 nodes as now reported clearly shows the relative merits of the proposed method and we believe nothing is lost by omitting simulated results for the larger network.
> >
> > _8. Moreover, when it comes to the comparison of different methods, simple statistical baselines (for instance a feature vector based on degrees, overall modularity of nodes, clustering coefficient, etc.) could also be employed (if they are unsuitable, this should at least be discussed). I would anticipate that k-means based on network summary statistics might perform better than k-means using vectorised weights? It would also be interesting to see a simple graph kernel being employed here, for instance a graph kernel using a graph-level histogram; since the networks are not excessively large, adding such additional comparison partners should still be possible._
> >
> > The methods used for comparison have been modified in response to this (and other) comments.  We now include methods based on higher dimensional simplices, vectorized network summary statistics, and two graph kernels, including a graph-level histogram. The aforementioned new methods account for the graph-level representation. We retain edge weight clustering using k-means as that represents an extreme case (lambda = 0) of the proposed method.

---

> > > ### Author Response · Authors · 2021-11-23
> > > **Response to Reviewer NiBH [3/3]**
> > >
> > > _9. The same comments apply, mutatis mutandis, to the discussion of the brain network data set. Here, it would be imperative to show different results for different values of λ._
> > >
> > > Figure 6 in the revised manuscript now depicts individual subject and group performance as a function of lambda.  These results suggest that both geometric distance between edge weights and topological distance are important.  However, the performance is not highly sensitive to the choice of lambda in the vicinity of lambda = 0.5, representing equal weighting of distances based on topology and geometry.
> > >
> > > _10. Moreover, given the large standard deviations, it seems that almost all methods are performing similarly here; is there a reason for such a high variance in the results? This should be briefly discussed._
> > >
> > > Different subjects have brain networks that are significantly different in terms of signal-to-noise ratio, the number of nodes, and the number of brain networks in each underlying state (wake, sedated, and unresponsive). For instance, there is one subject whose data consists of only wake and sedated states, that is, there are no measurements for the unresponsive state.  The differences in the nature of the data for each subject results in significant variability in clustering performance across subjects.  Consequently, the standard deviations across the entire data set are large even though there are clear performance differences between methods for many individual subjects.  We have added a graphical depiction of the performance on a subject by subject basis (Figure 7) to support this interpretation. Note that there is significant variability in performance across subjects.  The proposed topological clustering approach consistently outperforms other methods on a subject by subject basis. For some subjects (B384, L514) multiple methods work well, while for others (L400) none of the methods effectively cluster the data.
> > >
> > > This issue is now briefly discussed in the Results portion of Section 5.
> > >
> > > _11. Out of curiosity: would such an experiment benefit from a more fuzzy definition of clusters? The method should potentially allow for this, I think, and this might result in better clusters, in particular when it comes to an unclear delineation between wake and sedated states._
> > >
> > > Fuzzy clustering is a very interesting idea but given the space limitations and uncertainty in the neuroscience about the degree to which wake and sedated states differ, we decided to defer investigation of this idea to future work when it can be addressed more carefully.
> > >
> > > _12. Typos, word-smithing, etc.
> > > Overall, the writing in this paper flows well and the language is clear. There are some cases in which articles are missing or a plural form might be employed, but this does not prevent a clear understanding of the text._
> > >
> > > We have proofread the revised paper to correct as many grammatical errors as possible.
> > >
> > > _Here are two examples of this (additional ones can be found throughout the text; since some of them boil down to personal preference, I will refrain from listing them):_
> > > * _'cycle structure is ubiquitous' --> 'cycles are ubiquitous'_
> > >
> > > The sentence is modified as suggested.
> > >
> > > * _'takes infinitely large amount of work' --> 'takes an infinitely large amount'_
> > >
> > > The article is added as suggested.

---

> > > > ### Comment · Reviewer_NiBH · 2021-11-29
> > > > **Thanks for the updates**
> > > >
> > > > Thanks for the excellent rebuttal and revisions of the paper; they addressed my concerns and I am happy to increase my score accordingly.

---

### Official Review · Reviewer_VKeM · 2021-11-03

**Correctness:** 3
**Technical Novelty And Significance:** 2
**Empirical Novelty And Significance:** 2
**Recommendation:** 6
**Confidence:** 4

**Main Review:**

The proposed method is definitely interesting and the provided scores in the experiments are impressive. Moreover, the algorithm is backed up by not-surprising yet nice theoretical guarantees. However, I also think that the writing and the experiments could be improved.

In particular, I also have the following comments:

---Section 1, 2nd paragraph: it would be good to add a reference to the PersLay paper (http://proceedings.mlr.press/v108/carriere20a/carriere20a.pdf), which also provides a standard way to use (extended) persistence to characterize graphs.

---Section 1, 3rd paragraph: it would be good to add a reference to the entropic regularization for persistence diagrams (https://papers.nips.cc/paper/2018/hash/b58f7d184743106a8a66028b7a28937c-Abstract.html), which also contains a standard approach for approximating topological distances.

---Section 1, 4th paragraph: "by projecting the persistence barcodes into one dimension", it is a bit incorrect, as the proposed approach is not really about projecting 2D barcodes onto lines. The fact that the barcodes are 1D only comes from the data (i.e., graphs) on which they are computed. This should be corrected as otherwise it is confusing with actual projection techniques, such as, e.g., the sliced Wasserstein distance and kernel.

---Section 2.2: "there are |V|-1 connected components": in which graph? G_{-\infty} has one connected components and G_{\infty} has |V| connected components, so it is unclear to what space this sentence applies to.

---Section 4: the authors compare their method to k-means, but it is unclear on which data k-means is actually run. Is it over the adjacency matrices treated as vectors? But then, how is this different from the authors method applied with lambda=0? Similarly, isn't k-medoids with the bottleneck distance essentially the same than the authors method applied with lambda=1?

---Section 4: all in all, the authors method simply separates the filtration values into two groups based on 0- and 1-dimensional persistent homology, and then compare graphs with the Euclidean distance computed on the sorted groups. So I really wonder (and I think this should be included in the baseline) if the improvements in score could be obtained by simply adding the Euclidean distance on the whole sorted filtration values (without running persistent homology).

---It is always a bit suspicious to see p-values exactly equal to zero as in Table 1 (b). 100 iterations seems a bit low, maybe this number should be increased on a few thousands on these data sets.

---The improvement on the scores is quite impressive, but on the other hand the baseline seems a bit naive. Since the number of networks is not so large in the experiments, I strongly suggest adding fancier methods in the baseline, such as, e.g., kernel methods or neural nets. Indeed, there is quite a lot of graph kernels (https://arxiv.org/pdf/2011.03854.pdf) and graph neural networks (https://arxiv.org/pdf/1812.08434.pdf) in the literature, some of them performing quite well in applications, so it would nice to add some more comparisons.

---The authors distance is only presented on graphs with same number of nodes. Since their method extends naturally to graphs with different sizes (even though running time would probably be worse), it would be nice to discuss this at some point in the article.

[Post rebuttal comment]: the rebuttal was nicely done, and I think that the paper is now good enough for publication.

**Summary Of The Paper:**

In this article, the authors propose a way to characterize graphs using topology and persistent homology. Indeed, graphs represented with adjacency matrices can be filtered using the corresponding edge weights, and the resulting persistence diagrams can then be used for encoding the topological structures of the graph. Moreover, since the topology of graphs can be easily controlled (the only features they have are connected components and loops), their persistence diagrams are simple enough so that matching and distances between them can be computed efficiently. However, since topology alone can miss important information, the authors suggest to combine this topological distance with a more standard distance, namely the Frobenius norm between the adjacency matrices themselves. Then, they show how to use these distances in an expectation-maximization-like algorithm, using some theoretical computation guarantees of the Fréchet mean associated to their distances. Finally, they provide experiments in which their procedure compares favorably to competitors on a few synthetic and real-world graph classification tasks.

**Summary Of The Review:**

Overall I think that the technique proposed in the article is really promising, but that experimental results should be developed a little more before getting accepted.

---

> ### Author Response · Authors · 2021-11-23
> **Response to Reviewer VKeM [1/2]**
>
> We thank the reviewer for the detailed, thoughtful, and constructive review.  The reviewer’s suggestions are consistent with the other reviews.  We have addressed each recommendation as described below. The revisions have significantly improved the manuscript.
>
> The reviewers’ comments are numbered and summarized in _italics_ below. The response is in ordinary font.
>
> _1. ---Section 1, 2nd paragraph: add a reference to the PersLay paper_
>
> Done. We have also added an additional sentence on this general issue at the end of Section 2.1.
>
> _2. ---Section 1, 3rd paragraph: add a reference to the entropic regularization for persistence diagrams_
>
> Done.
>
> _3. ---Section 1, 4th paragraph: "by projecting the persistence barcodes into one dimension", it is a bit incorrect, as the proposed approach is not really about projecting 2D barcodes onto lines…._
>
> Please see the fourth paragraph of the introduction. The first sentence now clarifies that the persistence barcodes for network graphs are inherently one dimensional, which leads to simplified computation.  The title of Section 2 is also changed to reflect this perspective.
>
> *4. ---Section 2.2: "there are |V|-1 connected components": in which graph? G_{-\infty} has one connected components and G_{\infty} has |V| connected components, so it is unclear to what space this sentence applies to.*
>
> The wording in the second paragraph of Section 2.2 has been changed to clarify the intended meaning.  Note that |V|-1 edge weights out of |V|(|V|-1)/2 edges in the complete graph G_{-\infty} correspond to birth values of connected components.
>
> _5. ---Section 4: the authors compare their method to k-means, but it is unclear on which data k-means is actually run. Is it over the adjacency matrices treated as vectors? But then, how is this different from the author's method applied with lambda=0? Similarly, isn't k-medoids with the bottleneck distance essentially the same as the authors method applied with lambda=1?_
>
> Yes, k-means is run by vectorizing the adjacency matrices and is equivalent to the proposed topological clustering method based on d_{net} for the special case when lambda = 0. Note that when lambda = 0, the network dissimilarity metric d_{net} defined in Eq. (4) depends only on the geometric distance between the edge weights. So we instead term this method “edge weight” clustering in the revision. This extreme case of lambda = 0 does not use topological information about the networks and thus performs relatively poorly.
>
> Note that the case of lambda = 1 is based on the Wasserstein distance from the topological centroid and thus is not equivalent to k-medoids with bottleneck distance. Reviewer FgNK noted that use of bottleneck distance is inappropriate in this comparison, so we removed bottleneck distance from the revised manuscript. In its place we have introduced clustering based on Wasserstein distance between barcodes constructed by the previously proposed Rips filtration (Ghrist, 2018, Barcodes: The persistent topology of data).  This addition helps address the reviewer’s point above by contrasting the inherent one-dimensional nature of the barcodes for the proposed approach with the two-dimensional nature of the barcodes in other approaches.
>
> _6. ---Section 4: all in all, the authors method simply separates the filtration values into two groups based on 0- and 1-dimensional persistent homology, and then compares graphs with the Euclidean distance computed on the sorted groups. So I really wonder (and I think this should be included in the baseline) if the improvements in score could be obtained by simply adding the Euclidean distance on the whole sorted filtration values (without running persistent homology)._
>
> Thank you for the suggestion. We have added the case recommended by the reviewer where the edge weights are sorted and Euclidean distance between sorted edge weights is used as the distance metric. This approach performs nearly identically to the proposed approach on the simulated networks, most likely because these networks have relatively densely connected structure. Consequently, there are many more deaths of cycles than births of connected components, and the separation of a small portion of edge weights has limited impact on the overall topological distance. However, the effect of the separation is expected to increase on sparse networks with fewer cycles in the structure.
>
> _7. ---It is always a bit suspicious to see p-values exactly equal to zero as in Table 1 (b). 100 iterations seems a bit low, maybe this number should be increased to a few thousands on these data sets._
>
> The p-values were not exactly zero but had rounded to zero.  We avoid this confusion in the revision by reporting them as “<0.001”.  We did use several thousand iterations and found no meaningful difference in the results.

---

> > ### Author Response · Authors · 2021-11-23
> > **Response to Reviewer VKeM [2/2]**
> >
> > _8. ---The improvement on the scores is quite impressive, but on the other hand the baseline seems a bit naive. Since the number of networks is not so large in the experiments, I strongly suggest adding fancier methods in the baseline, such as, e.g., kernel methods or neural nets. Indeed, there is quite a lot of graph kernels (https://arxiv.org/pdf/2011.03854.pdf) and graph neural networks (https://arxiv.org/pdf/1812.08434.pdf) in the literature, some of them performing quite well in applications, so it would nice to add some more comparisons._
> >
> > Thank you for the suggestion. The revision now includes comparisons to several more sophisticated methods that had been proposed previously, including two graph kernel based methods, one graph theory based method and two Wasserstein distance based methods. While many of these methods perform very well with very strong topological structure (r = 0.9), their performance suffers as the topological structure becomes more noisy.
> >
> > Note that because some of these additional methods are extremely computational intensive and we added simulations showing performance as a function of the (effective) topology to noise level, there was not sufficient time in the revision period to obtain results with the new methods for the larger simulated network (120 nodes) provided in the original submission.  However, the simulation study with 60 nodes as now reported clearly shows the relative merits of the proposed method and we believe nothing is lost by omitting simulated results for the larger network.
> >
> > _9. ---The author's distance is only presented on graphs with the same number of nodes. Since their method extends naturally to graphs with different sizes (even though running time would probably be worse), it would be nice to discuss this at some point in the article._
> >
> > We have modified the organization of the paragraphs immediately prior to and after Lemma 1 to explicitly note that the lambda = 1 case may be applied to graphs with different sizes and cite a relevant reference. Analysis of clustering networks of different sizes is an interesting problem, but is beyond the scope of the present manuscript given the space limitations.

---

> > > ### Comment · Reviewer_VKeM · 2021-11-29
> > > **Response to authors**
> > >
> > > Thanks for taking my comments into account. I find the rebuttal well done so I increased my score.

---

### Public Comment · ~Yuri_Smirnov1 · 2021-11-15
**Missing citation**

A comment on missing citation. Since you are using the persistence barcodes of complexes, here is the reference where they were first introduced, under the name of canonical forms : Barannikov, S.(1994) "The Framed Morse Complex and its Invariants", Advances in Soviet Mathematics, 21: 93–115. Also, the computation of persistence barcodes is based on the algorithm described in section 2.1 of this paper.

---

> ### Author Response · Authors · 2021-11-23
> **Response to Yuri Smirnov**
>
> Thank you. The reference is cited in the revised manuscript.

---

### Decision · Program_Chairs · 2022-01-20

**Decision:**

Accept (Poster)

**Comment:**

Initially, we had some borderline scores for this paper. After the (indeed very convincing!) rebuttal and a the end of the discussion phase, however, all reviewers agreed that this is a very solid piece of work, with significant methodological and practical contributions. I fully share this positive impression of the paper!.